# Chemical combinations potentiate human pluripotent stem cell-derived 3D pancreatic progenitor clusters toward functional β cells

Haisong Liu[1,5], Ronghui Li[1,5], Hsin-Kai Liao[1,5], Zheying Min[2], Chao Wang[1], Yang Yu [1,2], Lei Shi[1], Jiameng Dan[1], Alberto Hayek[3], Llanos Martinez Martinez [4], Estrella Nuñez Delicado [4] & Juan Carlos Izpisua Belmonte [1✉]

Human pluripotent stem cell (hPSC)-derived pancreatic β cells are an attractive cell source for treating diabetes. However, current derivation methods remain inefficient, heterogeneous, and cell line dependent. To address these issues, we first devised a strategy to efficiently cluster hPSC-derived pancreatic progenitors into 3D structures. Through a systematic study, we discovered 10 chemicals that not only retain the pancreatic progenitors in 3D clusters but also enhance their potentiality towards NKX6.1+/INS+ β cells. We further systematically screened signaling pathway modulators in the three steps from pancreatic progenitors toward β cells. The implementation of all these strategies and chemical combinations resulted in generating β cells from different sources of hPSCs with high efficiency. The derived β cells are functional and can reverse hyperglycemia in mice within two weeks. Our protocol provides a robust platform for studying human β cells and developing hPSC-derived β cells for cell replacement therapy.

[1] Gene Expression Laboratory, The Salk Institute for Biological Studies, La Jolla, California, USA. [2] Beijing Key Laboratory of Reproductive Endocrinology and Assisted Reproductive Technology and Key Laboratory of Assisted Reproduction, Ministry of Education, Center for Reproductive Medicine, Department of Obstetrics and Gynecology, Peking University Third Hospital, Beijing, China. [3] Department of Pediatrics, UCSD-Medical School, La Jolla, California, USA. [4] Universidad Católica San Antonio de Murcia, Murcia, Spain. [5] These authors contributed equally: Haisong Liu, Ronghui Li, Hsin-Kai Liao. ✉email: belmonte@salk.edu

Remarkable progress has been made in the past decade in the guided differentiation of human pluripotent stem cells (hPSCs) into pancreatic β-cells, with the goal of using these cells for cell replacement therapy to cure type I diabetes[1–11]. These current protocols usually use various growth factors or small molecules to simulate the development of pancreatic β-cells and gradually induce hPSC through various intermediate states (definitive endoderm, primitive gut tube, posterior foregut, pancreatic progenitor (PP), and endocrine progenitor (EP)) to generate insulin-producing cells[1–11]. However, the application potential of current methodologies remains limited because of several major issues. First, the reported protocols for generating β-cells (marked by NKX6.1+/INS+) from hPSCs are still not efficient, ranging from ~10% to ~40%[2,6–8,11]. As a comparison, neuronal differentiation from hPSCs can reach up to ~80%[12]. To reduce the cost of β-cell production, greater efficiency is necessary. Second, hPSC-derived β-cell cultures are often heterogeneous and contain unwanted cell types that could possibly impede the maturation and function of β-cells by secreting interfering factors. Thus, purification and re-aggregation of hPSC-derived β-cells alleviated the negative influence and promoted β-cell maturation[13]. However, such a method is not efficient and increases the overall cost. Furthermore, heterogeneous cell cultures pose a risk of teratoma formation after transplantations[14,15]. Dual-hormonal GCG+/INS+ cells, which are considered immature α-cells, frequently emerge in β-cell differentiation cultures[6,9,16,17]. It remains a challenge to reduce the presence of these GCG+/INS+ cells. Third, current protocols have highly variable efficiencies depending on the origin of cell line manipulated. For example, Rezania et al.[7] generated ~40% NKX6.1+/INS+ β-cells from H1 human embryonic stem cells (hESCs), but only achieved ~10% efficiency when applying the same protocol on a human induced pluripotent stem cell (hiPSC) line[7]. Thus, current protocols are limited in their potential to generate patient-specific β-cells, because they might require intensive optimization for each cell line. We reasoned that these problems might be due to yet unknown mechanisms and/or the misregulation of some signaling pathways.

In this study, we perform systematic screenings on all the major steps during β-cell differentiation from hPSCs. We find that an extended culture of three-dimensional (3D) PP clusters with a cocktail of ten chemicals can boost their potential toward the derivation of β-cells. We also identify several factors or compounds or their combinations that have not been reported for inducing PPs into functional β (Fβ) cells, including forskolin (FSK) for the EP derivation stage, and ISX-9, G-1, 3-Deazaneplanocin A (Deza), ZM447439 (ZM), and CI-1033 for the last Fβ-cell stage. With all these improvements, we create a protocol that enables us to generate NKX6.1+/INS+ pancreatic β-cells from multiple hPSC lines with high efficiency (up to 82%) and minimal byproduct cells. The resulting β-cells are functional and capable to reverse hyperglycemia in diabetic model mice within 2 weeks.

## Results

### Efficient generation of PP 3D clusters

The generation of 3D clusters of PPs is considered a necessary first step in producing β-cells[7]. We initially followed the Rezania et al.[7] protocol (hereafter referred to as the R-protocol), as this protocol has been adopted by several other groups[16,18]. H1 hESCs were induced to form PPs (marked by PDX1+/NKX6.1+) in a step-wise mode and then assembled into 3D clusters according to the R-protocol. However, this resulted in PP production with only moderate efficiency (~30%). Moreover, the PP aggregates were not formed solidly and constant cell detachments were often observed. Therefore, cluster

size was difficult to be uniformly controlled. Consequently, subjecting these PP 3D clusters to further differentiation yielded only ~8–25% of NKX6.1+/INS+ β-cells. These results were not unexpected as other groups using the same procedures also achieved similar β-cell generation efficiencies[16,18]. We reasoned that the efficient generation of PP 3D clusters is critical for generating Fβ-cells. To improve this process, we first generated an hPSC reporter line with NKX6.1-NLS-GFP by homologous recombination (Fig. 1a, b). Sequences encoding green fluorescent protein (GFP) and a nuclear localization signal (NLS) were inserted in the 3′-end of the endogenous NKX6.1 gene in H1 hESCs. Restricting GFP to the nucleus increased local GFP fluorescence, enhancing the sensitivity so that we can readily distinguish live NKX6.1-GFP+ cells from NKX6.1-GFP− cells. Using this reporter cell line, we attempted to create a PP differentiation method based on a combination of the R-protocol[7], Nostro et al. protocol[19], and our previously published protocol[20]. Briefly, hPSCs were step-wise induced into definitive endoderm (by CHIR99021 and Activin-A), primitive gut tube (by Keratinocyte Growth Factor (KGF), Vitamin C (Vc), and dorsomorphin), posterior foregut (by KGF, Noggin, SANT1, retinoic acid (RA), and Vc), and PPs (by Epidermal Growth Factor (EGF), Vc, Nicotinamide (NICO), and Noggin). The resulting chemically defined protocol produced a monolayer of NKX6.1+/PDX1+ PPs with an efficiency of 84 ± 2% (n = 3) (Fig. 1c and Supplementary Fig. 1a–f). Of note, the dose of Activin-A at stage 1 was very critical and a decreasing gradient over the 3-day treatment resulted in most PPs at stage-4 (Supplementary Fig. 1e). In contrast, treating cells with constant low-dose or high-dose of Activin-A results in low NKX6.1+ cells at the PP stage (Supplementary Table 1). We also noticed that high-dose Activin-A severely reduced cell proliferation and resulted in suboptimal cell density. On the other hand, constant low-dose Activin-A generated cell culture with appropriate cell density but resulting in high cell heterogeneity at stage-4 (Supplementary Fig. 1e). These observations suggested that a balance between differentiation and proliferation is important for the efficient generation of PPs. We then improved aggregation methods to assemble these PPs into 3D clusters. To accomplish this, we used a V-bottom plate (Fig. 1d). Dissociated PP cells were added to V-bottom wells of a 96-well plate in the aggregation medium supplemented with the ROCK inhibitor Y27632, which increases the viability of dissociated cells. Approximately 0.2 million cells were pelleted in the bottom of the well by centrifugation. Compact 3D clusters were observed after overnight incubation and the clusters were then transferred to the air–liquid interface for further differentiation. Using this method, we robustly generated compact 3D clusters of PPs without significant loss of cells. These PP 3D clusters uniformly expressed high levels of NKX6.1-NLS-GFP, which is in sharp contrast to aggregates from undifferentiated hPSCs (Fig. 1e, f). To test the reproducibility of this protocol, we applied it to original wild-type H1 cells without the reporter and were able to achieve similar efficiency (81 ± 4%, n = 3) in generating PPs and assembling them into 3D clusters with uniform sizes (Fig. 1g). Taken together, we successfully established an efficient protocol for the generation of 3D clusters of PPs from hPSCs.

### Ten chemicals boost PPs' tendency for β-cell differentiation

We next sought to differentiate H1 hESC-derived 3D PP clusters into β-cells. We first applied the last three steps of the R-protocol to differentiate these PP clusters (we termed this combined protocol as Method 1). Briefly, the PP clusters were successively induced into pancreatic EPs (by SANT1, RA, Repsox, T3, and LDN193189 (LDN)), immature β-cells (iβ-cells) (by Repsox, T3, LDN, and γ-secretase inhibitor XX (GSIXX)), and maturing

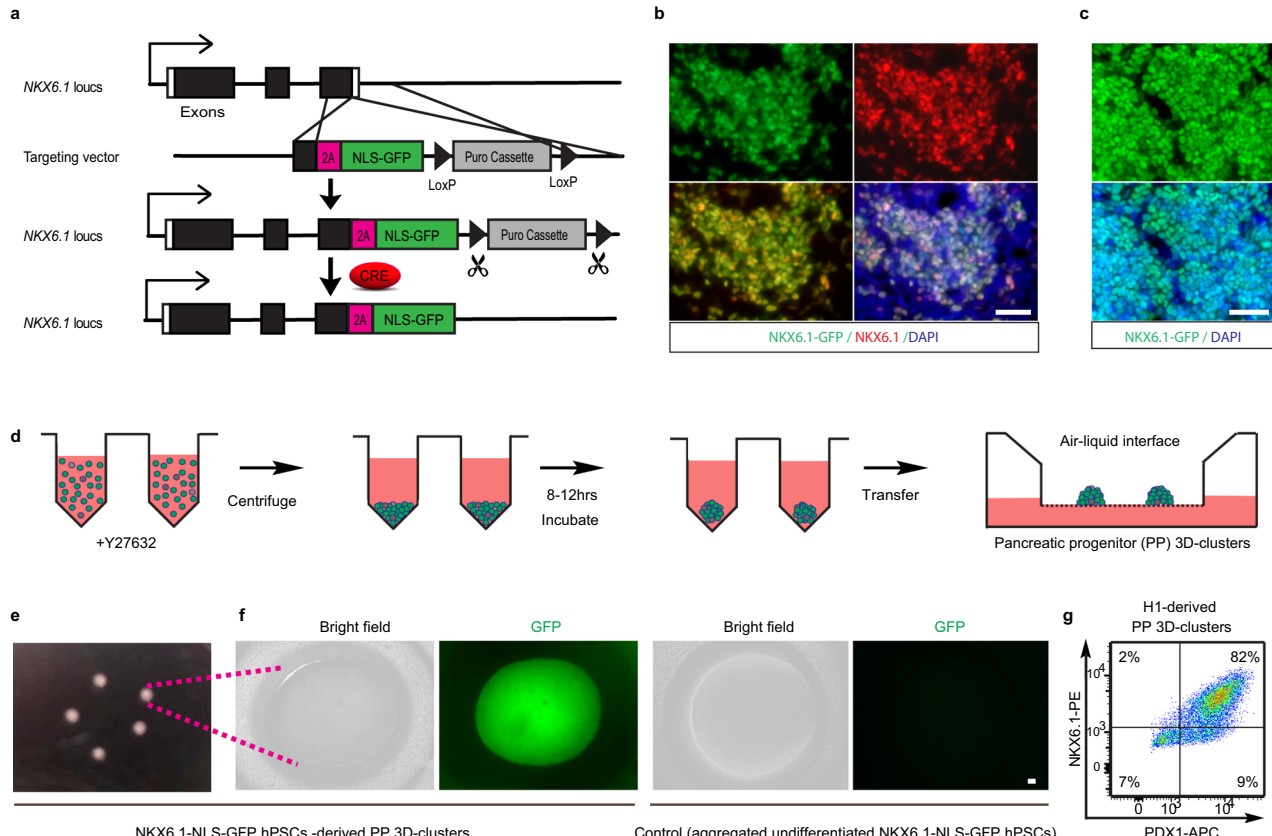

**Fig. 1 Efficiently generating pancreatic progenitor (PP) 3D clusters. a** A scheme showing the generation of the NKX6.1-NLS-GFP reporter hPSC line. Using a LoxP-flanked Puromycin (Puro) selection, a 2A-NLS-GFP was knocked in and fused in frame to the end of endogenous *NKX6.1* coding sequence. The Puro cassette was later removed by CRE excised. The cells will show nucleus-localized GFP expression once endogenous *NKX6.1* gene is activated. **b** Representative images of GFP and NKX6.1 immunofluorescence in NKX6.1-NLS-GFP hPSC-derived PPs showing the fidelity of the GFP reporter. **c** Representative images of GFP and nucleus for PPs derived from NKX6.1-NLS-GFP hPSCs using our improved protocol, showed the high induction efficiency of PPs. **d** Schematic graph of the strategy for assembling PPs into 3D clusters. The PPs were dissociated and re-aggregated by centrifuging in V-bottom 96-well plate to form cell pellets and then form 3D clusters after incubation. **e** Morphology of PP 3D clusters maintained on air–liquid interface. **f** Enlarged image of NKX6.1-NLS-GFP hPSC-derived PP 3D clusters showed the uniform expression of GFP (left panel); aggregated undifferentiated NKX6.1-NLS-GFP hPSCs showed no GFP expression (right panel). **g** Cell cytometry analysis showed the high percentage of PPs (NKX6.1+/PDX1+) existing in H1 cell-derived PP 3D clusters (*n* = 3). Scale bar, 100 μm. Images of **b**, **c**, **e**, and **f** are representative of three independent experiments, respectively.

β-cells (by Repsox, T3, *N*-acetyl cysteine, Trolox and AXL inhibitor). We found, however, despite that the production of ~56% INS+ cells, most of the INS+ cells also express GCG and only 14 ± 4% (*n* = 3) of total cells were β-cells (marked by NKX6.1+/INS+) (Fig. 2a, b and Supplementary Fig. 2a). The percentage of NKX6.1-expressing cells dropped from ~80% to ~19% after the last three stages of the R-protocol. The prevalent INS+/GCG+/NKX6.1− cells suggested that the last three steps of the R-protocol mainly induced the PP clusters into INS+/GCG+ bi-hormonal cells, rather than INS+/NKX6.1+ β-cells. As it has been reported that most INS+/NKX6.1− cells expressed GCG[17], and given the critical role of NKX6.1 for maintaining the functional state of β-cells[21], we hypothesized that the premature loss of NKX6.1 expression and/or PPs identity may have led to the INS+/GCG+/NKX6.1− cell fate. Therefore, we reasoned that maintaining cells in the PP stage with compact 3D structures until they were fully committed might be critical to enhance their propensity to become β-cells. To test this hypothesis, we treated 3D clusters of PPs with different combinations of factors, screening for conditions that retained NKX6.1 expression in the PP state without compromising the compaction of the 3D structures. In many of the conditions tested, NKX6.1 expression dropped significantly just 4 days after the formation of PP 3D clusters (Fig. 2c and Supplementary Fig. 2b), suggesting that

NKX6.1 expression could be lost prematurely under undesirable conditions. Only condition #13 could maintain NKX6.1 expression after 4 days of treatment (Fig. 2c, d and Supplementary Fig. 2b). Condition #13 contained ten chemicals, namely LDN (an inhibitor of BMP signaling), T3 (thyroid hormone), RA, SANT1 (an inhibitor of sonic hedgehog signaling), Repsox (an ALK5 inhibitor), ZnSO₄, (2S,5S)-(E,E)-8-(5-(4-(tri-fluoromethyl)phenyl)-2,4-pentadienoylamino) benzolactam (TPB, a protein kinase C activator), EGF, NICO (a form of vitamin B₃), and γ-aminobutyric acid (GABA, a neuro-transmitter). As condition #13 was used for treating PPs, we renamed this combination of ten chemical/factors as "PP-10C". Further characterization of PP-10C-treated PPs proved that 83 ± 5% (*n* = 3) cells expressed both PDX1 and NKX6.1, but not middle- or late-stage markers of endocrine cells, such as NGN3 or NEUROD1 (Fig. 2d–g). The results of terminal deox-ynucleotidyl transferase dUTP nick end labeling (TUNEL) staining (for detecting cell death) with NKX6.1 and 4′,6-diami-dino-2-phenylindole (DAPI) revealed very few dying/dead cells in PP-10C conditions (Supplementary Fig. 2b, condition #13). In contrast, we did observe more dying/dead cells in other conditions. Moreover, we found necrotic cores (with a lot of pervading DAPI stain, dying/dead cells, and cell debris) in some other conditions (Supplementary Fig. 2b, condition #4, #5, and #19).

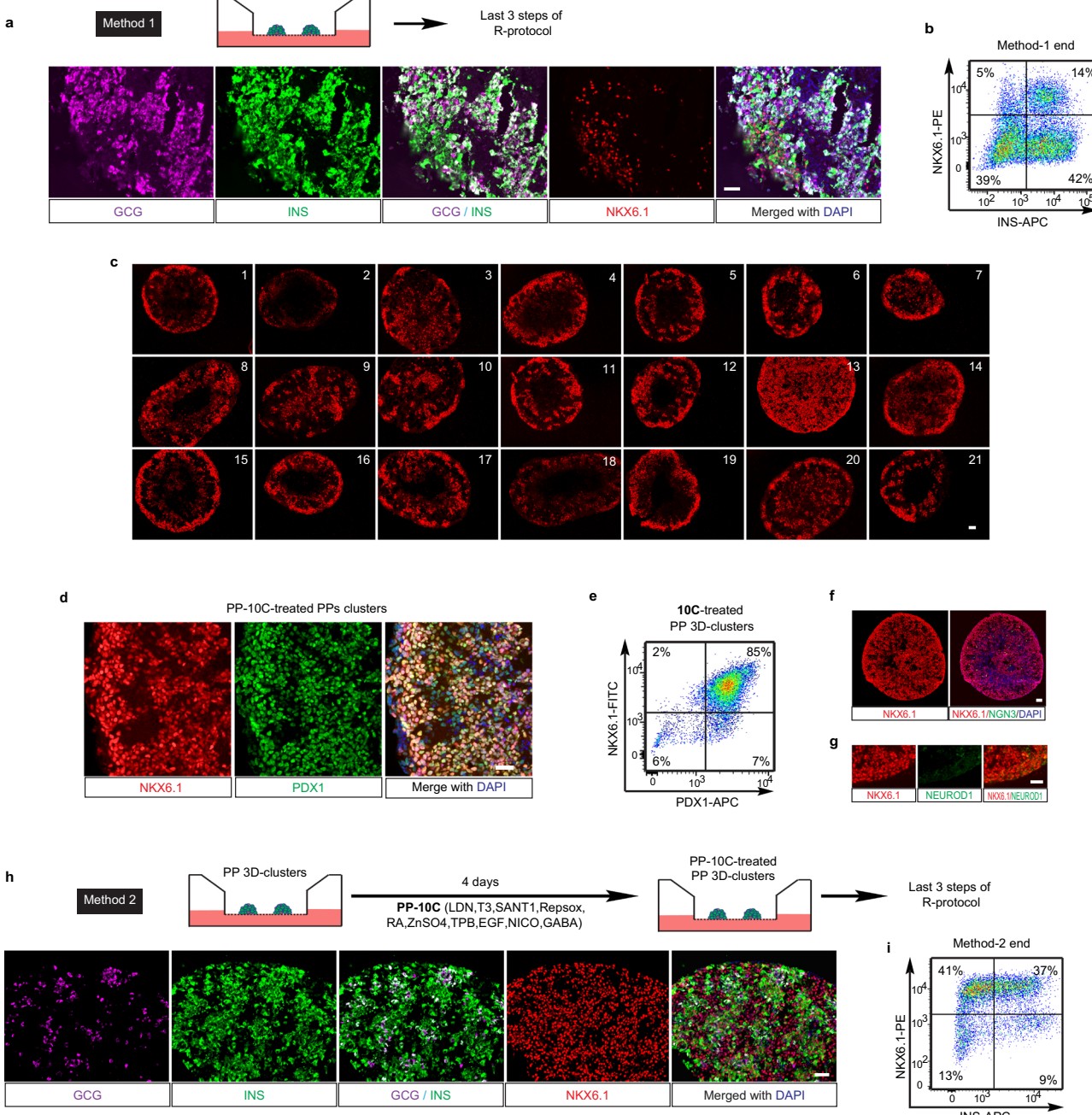

**Fig. 2 Ten chemicals poise pancreatic progenitors in 3D clusters. a, b** Method 1: H1 cell-derived PP 3D clusters were induced into β-cell stage, using the last three steps of R-protocol. Both immunofluorescent staining (**a**) and cell cytometry analysis (**b**) showed that end-stage cell culture comprised high percentage of INS+/NKX6.1− cells (mainly GCG+/INS+ cells) and a low number of β-cells (NKX6.1+/INS+). **c** H1 cell-derived PP 3D clusters were tested in 21 different conditions for 4 days (for detailed experimental design, see Supplementary Method). Section and staining showed condition #13 (10 chemicals (PP-10C)) is the best for maintaining NKX6.1 expression. **d–g** Immunostaining for NKX6.1 and PDX1 (**d**), NKX6.1 and NGN3 (**f**), NKX6.1 and NEUROD1 (**g**), and cell cytometry for NKX6.1 and PDX1 (**e**), in PP-10C-treated PP 3D clusters, indicating PP-10C can maintain PPs status with minor differentiation. **h, i** Method 2: H1 cell-derived PP 3D clusters were first treated by PP-10C and then induced into β-cell stage using the last three steps of the R-protocol. Both immunofluorescent staining (**h**) and cell cytometry analysis (**i**) showed that the end-stage cluster comprised higher percentage of β-cells (NKX6.1+/INS+) and reduced number of GCG+/INS+ cells. Nuclear DAPI staining is shown in blue. Scale bar, 100 μm. Source data are provided as a source data file. Images of **a, c, d, f–h** are representative of three independent experiments, respectively. n = 3 for cell cytometry data **b, e**, and **i**.

Together, this indicated that PP-10C not only efficiently maintained the undifferentiated status of PPs but also retained the viability of all cells in 3D clusters. We then tested the potential of these PP-10C-treated PP clusters to differentiate into β-cells. Strikingly, subjecting PP-10C-treated PPs to the last three steps of the R-protocol (we termed this entire experimental protocol as Method 2) produced 36 ± 6% (n = 3) NKX6.1+/INS+ β-cells (a ~3-fold increase over Method 1) and only ~9% INS+/NKX6.1− byproduct cells (a ~5-fold decrease over Method 1) (Fig. 2h, i). Thus, PP-10C treatment increased the potential of PPs in 3D clusters to differentiate into β-cells, while reducing their potential to become unwanted INS+/GCG+ byproduct cells.

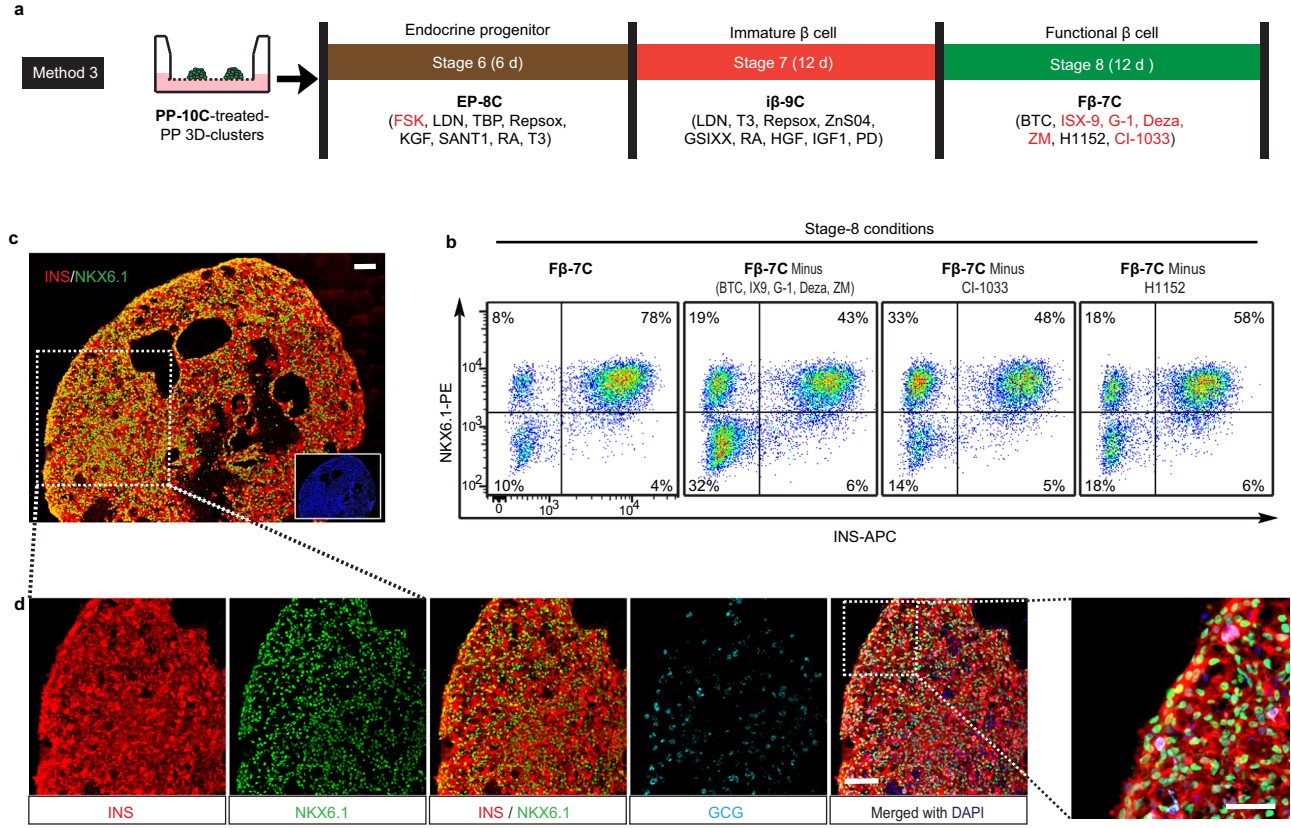

**Fig. 3 Systematic screening for the differentiation of pancreatic progenitors (PPs) into β-cells. a** A scheme of inducing PP-10C-treated PPs into β-cells using the improved protocol. There are extra five differentiation stages from hPSCs to PP-10C-treated PPs, so these three late stages are named as stage-6, -7, and -8, respectively. Chemicals/factors highlighted in red have not been reported in literatures for inducing hPSCs into β-cells. The chemicals/factors used in stage-6, -7, and -8 are termed as EP-8C, iβ-9C, and Fβ-7C, respectively. **b** Cell cytometry analysis of stage-8 cells showed Fβ-7C collectively induced (NKX6.1+/INS+) β-cells from earlier stages, while subtracting some factors from Fβ-7C at stage-8 reduced the efficiency (n = 5). **c, d** 10C-treated PPs were induced to β-cells according to the Method 3 shown in Fig. 3a, followed by immunostaining in the end-point cells, showing robust expression of INS, NKX6.1 but not GCG. Figure 3c was stitched by individual images. The inserted small image on the right bottom of Fig. 3c was the DAPI staining. Scale bar, 50 μm for the right panel of 2D and 100 μm for all others. Images of **c** and **d** are representative of five independent experiments, respectively.

**Identify chemical combinations to generate β-cells from PPs.** To further increase β-cell yield, we attempted to improve the step-wise procedure for differentiating PP-10C-treated PPs into EPs, then into iβ-cells, and finally Fβ-cells (Fig. 3a and Supplementary Fig. 3a). We named these last three steps stage-6, -7, and -8, respectively, as five previous stages were needed to differentiate hPSCs into PP-10C-treated PPs (Fig. 3a and Supplementary Fig. 3a). To search for potentially unknown signaling pathways involved in these differentiation stages, we generated a custom screening library comprising more than 100 chemicals/growth factors, which could modulate (activate or inhibit) most of the known developmental and differentiation-related signaling pathways (Supplementary Table 2). To this end, we systematically screened >2000 conditions using different combinations of chemicals/factors from the library. To improve stage-6, we initially used the classic EP markers, NKX6.1+/NEUROD1+, as the readout. We identified several conditions that efficiently generated NKX6.1+/NEUROD1+ cells from PP-10C-treated PPs (Supplementary Fig. 3b). However, we were not able to efficiently induce these NKX6.1+/NEUROD1+ cells into β-cells. This indicated that the expression of a few marker genes characteristic of an intermediate stage was not necessarily a good indicator of their potential to differentiate into β-cells. Thus, we designed a more stringent screen that instead used late-stage markers (NKX6.1+/INS+ for this case) as the readout, termed "late-stage readout strategy" (Supplementary Fig. 3c). Using this strategy, at

stage-6 (EP stage), we identified a combination of eight chemicals/factors (termed EP-8C), which gave rise to the highest percentage of NKX6.1+/INS+ cells after further differentiation (Fig. 3a and Supplementary Fig. 3a, c). The EP-8C combination was FSK (a cAMP pathway activator), Repsox (only 1 μM), LDN, TPB, KGF, SANT1, RA, and T3. Adding FSK and using a low concentration of Repsox was critical for this differentiation process. Retrospective analysis showed that EP-8C also efficiently generated NKX6.1+/NEUROD1+ cells at stage-6 (Supplementary Fig. 3d). We further improved stage-7 and -8, and the best conditions discovered are shown in Fig. 3a. Specifically, stage-7 (iβ-cell stage) required nine chemicals/factors (termed as iβ-9C), namely LDN, T3, Repsox, ZnSO4, GSIXX (a notch pathway inhibitor), RA, HGF, IGF1, and PD173074 (PD, a fibroblast growth factor (FGF) pathway inhibitor) (Fig. 3a and Supplementary Fig. 3e). Stage-8 (Fβ-cell stage) required seven chemicals/factors (termed as Fβ-7C), namely betacellulin (BTC), ISX-9 (a NEUROD1 inducer), G-1 (a G protein-coupled estrogen receptor agonist), Deza (a histone methyltransferase inhibitor), ZM (an aurora kinase inhibitor), H1152 (a ROCK-II inhibitor), and CI-1033 (a pan-ErbB inhibitor) (Fig. 3a, b and Supplementary Fig. 3a). Therefore, we established an efficient three-stage protocol for inducing PP-10C-treated PPs into β-cells (Fig. 3a and Supplementary Fig. 3a). For convenience, we refer to this experimental protocol as Method 3. To the best of our knowledge, EP-8C, iβ-9C, and Fβ-7C are all novel combinations of chemicals/

factors. Moreover, current literature suggests that some of these chemicals/factors (including FSK for stage-6, and ISX-9, G-1, Deza, ZM, and CI-1033 for stage-8) have not previously been used to differentiate hPSCs into β-cells (Fig. 3a, in red). Using Method 3, we generated NKX6.1+/INS+ β-cells with an efficiency of up to 82% (75 ± 6%, $n = 5$) (Fig. 3b and Supplementary Table 3). For stage-8, each Fβ-7C component enhanced the generation of β-cells, as removing any component compromised the production of NKX6.1+/INS+ β-cells (Fig. 3b). In agreement with the data from flow cytometry, immunostaining sections of stage-8 cell aggregates revealed robust expression of NKX6.1 and INS, and only a few cells that were dual-hormonal byproduct cells that expressed GCG and INS but not NKX6.1 (Fig. 3c, d and Supplementary Fig. 3f). In addition, C-peptide (a stoichiometric indicator of proinsulin processing) and PDX1 (a transcription factor expressed by mature β-cells) were also robustly expressed in the stage-8 clusters (Supplementary Fig. 4a). We have tested in total five cell lines (including H1, H1-NKX6.1-GFP, and three integration-free hiPSC lines (hiPSC1, hiPSC2, and hiPSC3)) for our protocol. We can routinely generate >60% β-cells from any of them (Supplementary Fig. 4b and Supplementary Table 3). In addition, our protocol is reproducible for the same cell line with low batch variations. These results suggest that our protocol would be suitable for many ES and high-quality iPS cell lines.

**β-Cells were functional both in vitro and in vivo.** Finally, we performed physiological tests on the stage-8 β-cells from Method 3 in vitro and in vivo. For examining glucose-sensing activity, an in vitro glucose-stimulated C-peptide secretion assay showed that the stage-8 cells responded to high-concentration (16.7 mM) glucose but not low-concentration (3.3 mM) glucose, and released human C-peptide (as normally seen from isolated human pancreatic islets) (Fig. 4a). The C-peptide levels induced by glucose from stage-8 cells were similar to primary human pancreatic islets. In contrast, immature stage-7 cells did not respond to high-concentration (16.7 mM) glucose (Fig. 4a). Stage-8 cells also responded to other secretagogues including Exendin-4 and KCl (Fig. 4b). In addition, we also measured the total insulin content in stage-8 cells, which is ~62 ng per 10,000 cells, a value comparable to that in primary human islets (Fig. 4c). Furthermore, to functionally test the stage-8 cells in vivo, we transplanted them under the kidney capsule of streptozotocin (STZ)-induced diabetic NOD-SCID-γ (NSG) mice. Diabetic mice that received these transplanted cells quickly recovered to normoglycemia (within ~2 weeks), whereas diabetic mice that received fibroblasts as a control maintained hyperglycemia (Fig. 4d). The in vivo glucose-stimulated C-peptide secretion assay also revealed robust secretion of human C-peptide in diabetic mice transplanted with the stage-8 cells. In these mice, about two times more C-peptide was detected after glucose stimulation compared to the fasting state (before glucose stimulus) (Fig. 4e). To test the safety of these β-cells, we transplanted cells from different differentiation stages (including, undifferentiated H1 hPSCs, stage-4, -7, and -8 cells) into normal NSG mice. As shown in Fig. 4f, in contrast to the massive teratoma formed by undifferentiated H1 hPSCs after 15 weeks, the stage-8 and stage-7 cells did not form teratomas even after 20 weeks post transplantation. Stage-4 cells formed some cyst-like structures (Fig. 4f). Analysis of the stage-8 cell-derived engraftments showed robust expression of INS (Supplementary Fig. 4c). Collectively, the β-cells generated using our protocol exhibited functionality both in vitro and in vivo.

## Discussion

The efficient generation of Fβ-cells from hPSCs for the treatment of diabetes has been a persistent challenge in the field of

regenerative medicine. Current methods generate β-cells with suboptimal efficiency and the effectiveness of these protocols varies when applied to different cell lines. Here we report the establishment of a robust chemical recipe for the highly efficient production of NKX6.1+/INS+ β-cells from hPSCs (Fig. 5). Our method incorporates several inventions, including (1) a chemically defined protocol for the efficient generation of PPs, (2) an improved method for assembling PPs into 3D clusters, (3) a PP-10C that maintains 3D-PPs status and enhances their potential to differentiate into β-cells (rather than unwanted byproducts such as GCG+/INS+ cells), and (4) a three-step differentiation protocol (with combinations of signaling pathway regulators for each step), which efficiently converts PP-10C-treated 3D-PPs into Fβ-cells. As our method significantly increases the efficiency of generating β-cells from multiple cell lines and reduces unwanted cellular byproducts, our discoveries will promote both basic research and clinical translation of hPSC-derived β-cells.

One guiding principle concerning the differentiation of hPSC into a specific cell type of interest is that cells must be induced to transiently progress through several progenitor states. Our study highlights the importance of poising certain 3D-progenitor states during the differentiation process. These steps are beneficial likely because the transient appearance of a progenitor state does not entail the full acquisition of the cell state, and the premature loss of a progenitor state often leads to the generation of undesirable cellular byproducts in the downstream steps. We believed that our strategy of poising progenitor cells during the differentiation process is applicable for generating a range of cell types from hPSCs and warrants further study. Another useful strategy we implemented in our screening process was the use of late-stage gene markers as the readout for the intermediate steps (namely, the late-stage readout strategy). Although this strategy is relatively labor-intensive, the resulting conditions are generally more reliable and reproducible than those identified solely based on stage-specific gene expression marker(s). This strategy could also be adapted for generating other cell types from hPSCs.

Our study also reveals the importance of several signaling pathway modulators in promoting the differentiation of hPSCs into β-cells. Of these modulators, (1) FSK has been reported to facilitate the process of epithelial-to-mesenchymal transition (EMT) in several cell systems and, thus, might promote PP into EP at Stage-6 through inducing EMT (EMT is a critical step during EP specification)[22–25]; (2) ZM, an aurora kinase inhibitor, might promote terminal cell maturation at Stage-8 by facilitating cell cycle exit[26,27]; (3) ISX-9, might enhance hPSCs-derived β-cell function and maintain their identity at stage-8 by inducing and maintaining NEUROD1 and INS expression in these cells[28]; (4) G-1, a G protein-coupled estrogen receptor agonist, possibly promotes β-cell maturation at stage-8 by modulating estrogen receptor-related pathways as hPSC-derived PPs mature more quickly in female than in male mice[29]. Further investigation is warranted to determine how exactly these signaling pathways exert their effects during the differentiation process.

## Methods

**Mice**. NOD-scid IL2-γ-null (NSG) mice were from the Jackson Laboratory. Mice were housed in 12 h light/12 h dark cycle, with temperatures of 65–75 °F and 40–60% humidity. All procedures related to animals were performed in accordance with the ethical guidelines of the Salk Institute for Biological Studies. Animal protocols were reviewed and approved by the Salk Institute Institutional Animal Care and Use Committee before any experiments were performed. Six- to eight-week-old male NSG mice were used for surgery.

**hESC differentiation**. A step-by-step protocol describing the differentiation protocol can be found at Protocol Exchange[30]. hESCs are maintained as feeder-free in mTeSR™1 (STEMCELL Technologies, Inc.) according to the manufacturer's instructions. Before differentiation, adherent hESCs were rinsed with phosphate-

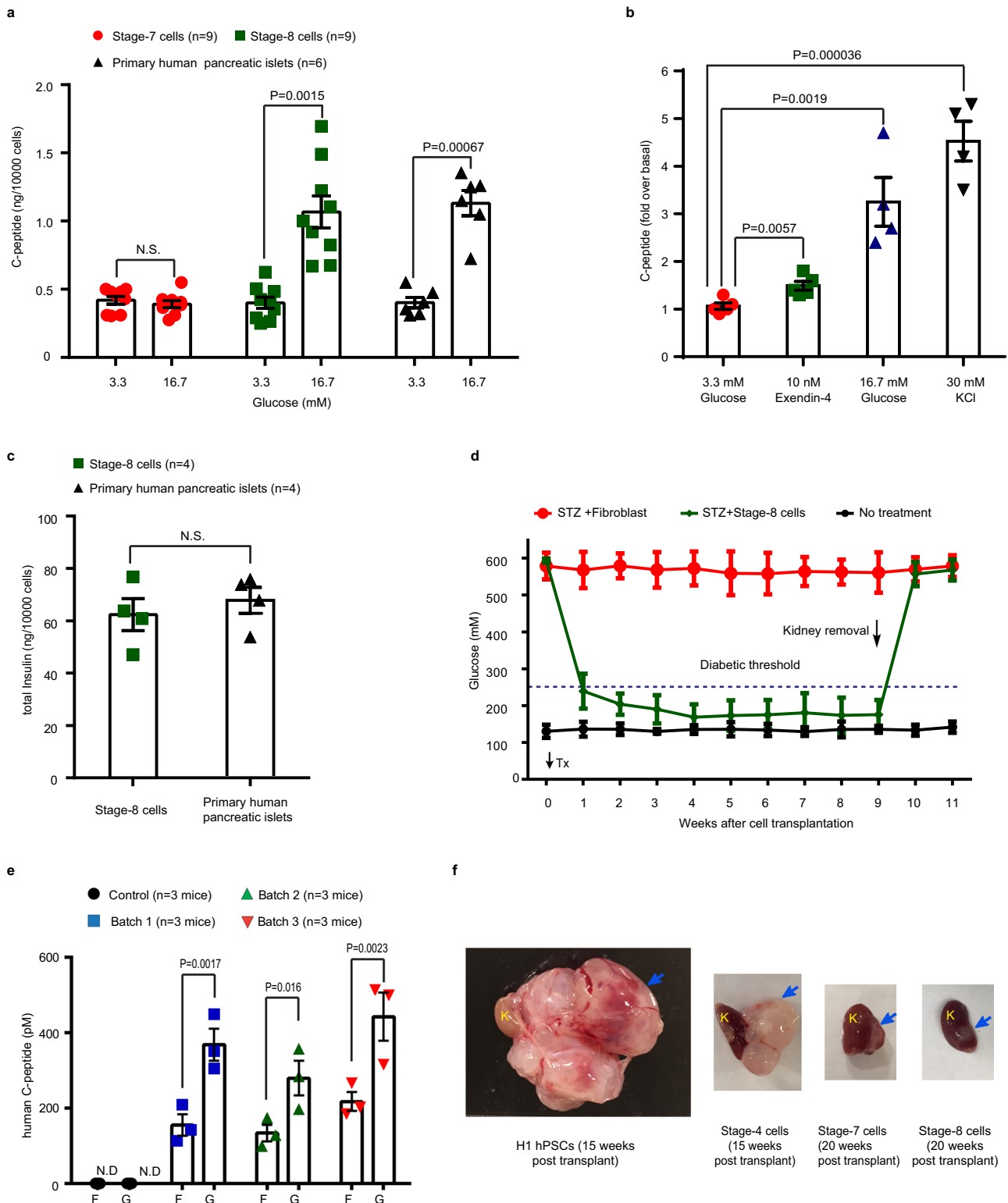

f

H1 hPSCs (15 weeks post transplant)

Stage-4 cells (15 weeks post transplant)

Stage-7 cells (20 weeks post transplant)

Stage-8 cells (20 weeks post transplant)

buffered saline (PBS) and then incubated with Accutase (Millipore) for 6–8 min at 37 °C. Dissociated single cells were rinsed twice with Dulbecco's modified Eagle's medium (DMEM)/F12 and spun at 300 × *g* for 3 min. The resulting cells were re-suspended in mTeSR™1, which was supplied with 10 μM Y27632 (Sigma-Aldrich), and seeded on 1:30 diluted Matrigel (BD Biosciences)-coated dishes at a density of 55,000 cells/cm². The next day, the medium was exchanged for mTeSR™1 and maintained for one more day prior to differentiation initiation. The H1 hESC line was obtained from WiCell Research Institute, Inc. (Madison, WI) and iPS cell lines were obtained from the Stem Cell Core of the Salk Institute.

*Stage 1: Definitive endoderm (3 days).* After PBS rinse, undifferentiated hESCs (at ~90% confluence) were exposed to differentiation medium as follows: day1—MS12 medium supplied with 3 μM CHIR99021 and 115 ng/ml Activin-A (ActA, Peprotech); day2—MS12 medium with 0.3 μM CHIR99021 and 110 ng/ml Activin-A; day3—MS12 medium with 100 ng/ml Activin-A. Each 800 ml MS12 medium was made with 744 ml MCDB131 medium (ThermoFisher), 3.2 ml 45% glucose (Sigma), 20 ml 20% fat–acid-free bovine serum albumin (BSA), 8 ml Pen/Strep (100×, Gibco), 8 ml Glutamax (100×, Gibco) and 16 ml 7.5% sodium bicarbonate (Gibco).

**Fig. 4 β-Cells were functional both in vitro and in vivo. a** Human C-peptide secretion from stage-7 (red filled circle), stage-8 (green box) cells, and primary human pancreatic islets (black triangle) in response to low (3.3 mM) and high (16.7 mM) glucose concentrations under static conditions. **b** Human C-peptide secretion from stage-8 cells in response to low glucose (3.3 mM, $n = 5$, red filled circle), Exendin-4 (with 3.3 mM glucose, $n = 5$, green box), high glucose (16.7 mM, $n = 4$, blue triangle), and 30 mM KCl ($n = 4$, black triangle). **c** Total insulin content of stage-8 cells (green box) and human islets (black triangle). **d** Stage-8 cells reversed the hyperglycemia in STZ-induced diabetic NSG mice. STZ was administrated ~5 weeks before cell transplantation. Tx, transplantation. $N = 4$ for experimental group and $n = 3$ for control groups. Transplanted fibroblasts were used as control. **e** Human C-peptide levels were measured after an overnight fasting and 60 min following an i.p. glucose bolus at 5 weeks post transplant of stage-8 end-point cells ($n = 3$ mice for each differentiation-and-transplantation batch). **f** Morphology of engraftments derived from different cells after transplantation in normal NSG mice, including stage-4, -7, and -8 cells, and undifferentiated H1 hPSCs. The blue arrows indicate engraftments. K, kidney. Data are presented as mean ± SEM for **a**, **b**, **d**, and **e**, and mean ± SD for **c**, respectively. $P$-values were determined by t-tests (two-sided) for **a**, **b**, **d**, and **e**. N.D., not detectable. N.S., not significant. The estimated purity of primary human pancreatic islets was about 70%. Source data are provided as a source data file. All the experiments repeated in **a**–**d** are biologically independent. Images of **f** are representative of three independent experiments.

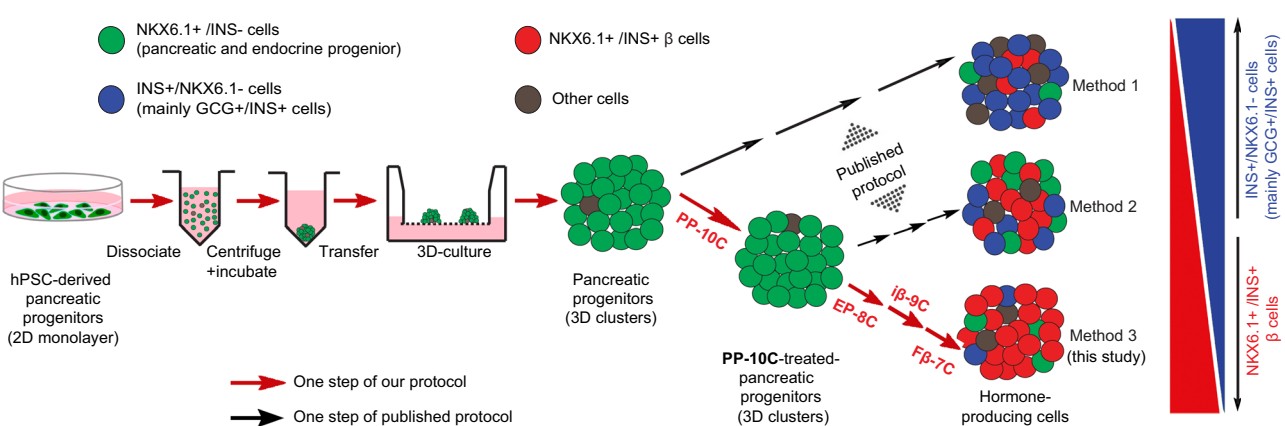

**Fig. 5 A schematic diagram summarizing the discoveries revealed in this study.** HPSC-derived PPs are dissociated and re-aggregated into PP 3D clusters by centrifuge and incubation in a V-shaped-bottom microplate and then transferred to the air–liquid interface. The PPs in 3D clusters mainly differentiate into β-cells after they are successively treated by PP-10C, EP-8C, iβ-9C, and Fβ-7C. However, if other suboptimal conditions are employed, the yield of β-cells is significantly reduced, whereas that of the dual-hormonal GCG+/INS+ cells increase.

*Stage 2: Primitive gut tube (3 days)*. After PBS rinse, cells are exposed to MS12 medium with KGF (Peprotech, 50 ng/ml), B27 (Gibco, 100×), Vc (0.25 mM, Sigma), and dorsomorphin (0.75 μM). Cells were fed with fresh medium daily.

*Stage 3: Posterior foregut (3 days)*. After PBS rinse, cells are exposed to DMEM with B27 (100×), RA (2 μM, Sigma), Noggin (100 ng/ml, Peprotech), SANT1 (0.25 μM, Sigma), Vc (0.25 mM, Sigma), Pen/Strep (100×), and Glutamax (100×). Cells are fed with fresh medium every other day.

*Stage-4: PP (3–4 days)*. After PBS rinse, cells are exposed to DMEM with B27 (100×), EGF (100 ng/ml, Peprotech) + NICO (Sigma, N0636, 10 mM) + Noggin (100 ng/ml) + Vc (0.25 mM). Cells are fed with fresh medium every other day.

   V-bottom plate-based aggregation: stage-4 cells were rinsed with PBS and then incubated with Accutase (Millipore) for 10–15 min at 37 °C. Dissociated single cells were rinsed twice with DMEM/F12 and spun at $300 \times g$ for 3 min. The resulting cells were re-suspended in aggregation medium (5a-Medium supplied with 10 μM Y27632). Cell solution with 0.1–0.4 (according to experimental design) million cells was added into each well of V-bottom 96-well plate, followed by a spun at $300 \times g$ for 3 min. The plate was put into 37 °C incubator for 8–12 h to form clusters. 5a-Medium is made of V4b-Medium + heparin (Sigma, H3149, 10 μg/ml) + ZnSO₄ (10 μM, Sigma, Z0251) + LDN (Selleck Chemical, S2618, 100 nM) + T3 (1 μM, Sigma, T6397) + RA (0.05 μM) + SANT1 (0.25 μM; Tocris) + GABA (1 mM, Sigma) + human EGF (100 ng/ml, Peprotech) + NICO (10 mM) + Vc (0.25 mM). V4b-Medium (800 ml) is made of 340 ml MCDB 131 medium + 170 ml F12 medium + 170 ml KO-DMEM medium + 3.9 ml Glucose (45%, Sigma) + 80 ml 20% FF-BSA + 16 ml Sodium Bicarbonate (7.5%) + 4 ml ITX + 8 ml Pen/Strep + 8 ml Glutamax; all items are from Thermo Fisher Scientific unless indicated.

*Stage-5: PP-10C-treated PP (4 days)*. PP clusters assembled in V-bottom 96-well plate were collected and rinsed twice using DF12 medium, and loaded on 6-well air–liquid interface transwell (Corning, 07200170). About 1.3 ml stage-5 medium was added for each well. Stage-5 medium is made of 5a-Medium + TPB (a protein kinase C activator, EMD Millipore, 565740; 100 nM) + Repsox (10 μM, 374210, Thermo Fisher Scientific). Cells are fed with fresh medium every other day.

*Stage-6: EP (6 days)*. Clusters were rinsed in DF12 and transferred to new transwell. About 1.3 ml stage-6 medium was added for each well. Cells are fed with fresh medium every other day. Stage-6 medium was made of B26-medium + FSK (10 μM) + T3 (1 μM). B26-medium = DMEM + B27 (100×) + LDN (500 nM) + TPB (30 nM) + Repsox (1 μM) + KGF (25 ng/ml) + SANT1 (0.25 μM) + RA (0.05 μM) + PS (100×).

*Stage-7: iβ-cells (12 days)*. Clusters were rinsed in DF12 and transferred to new transwell. About 1.3 ml stage-7 medium was added for each well. Cells are fed with fresh medium every other day. Stage-7 medium is made of 4#-medium supplemented with GSIXX (100 nM, only for day1–6) + RA (0.05 μM, only for day1–6) + HGF (50 ng/ml, only for day1–6) + IGF1 (50 ng/ml, only for day1–6) + FGF inhibitor PD (0.1 μM, only for day3–6). 4#-medium = V4b-medium + ZnSO₄ (10 μM) + Heparin (10 μg/ml) + LDN (100 nM) + T3 (1 μM) + Repsox (10 μM).

*Stage-8: Fβ-cells (12 days)*. Clusters were rinsed in DF12 and transferred to a new transwell. About 1.3 ml stage-8 medium was added for each well. Cells were fed with fresh medium every other day. Stage-8 medium was made of M18#-medium, supplemented with seven chemicals/factors (termed as Fβ-7C, including BTC (10 ng/ml, only used for day 7–12), ISX-9 (a NEUROD1 inducer, 10 μM), G-1 (a G protein-coupled estrogen receptor agonist, 1 μM), Deza (a histone methyl-transferase inhibitor, 1 μM), ZM (an aurora kinase inhibitor, 2.5 μM), H1152 (a ROCK inhibitor, 10 μM; only used for day 1–6), and CI-1033 (a pan-ErbB inhibitor, 1 μM, only used for day 7–12). M18 medium (5.58 mM glucose) = MCD131 (ThermoFisher, 340 ml for each 800 ml total medium) + F12 (ThermoFisher, 170 ml for each 800 ml total medium) + DMEM-no Glucose (ThermoFisher, 170 ml for each 800 ml total medium) + fatty acid-free BSA (final 2%, EMD Millipore, 126575) + 7.5% sodium bicarbonate (12.8 ml for total 800 ml medium) + RPMI 1640 Vitamins Solution (100×, Sigma-Aldrich, R7256) + ITX (200×) + sodium pyruvate (200×) + NEAA (200×) + PS (100×) + Glutamax (100×) + 45% glucose (Sigma, 2500×) + Heparin (10 μg/ml) + Lipid (2000×, Thermo Fisher Scientific, 11905031) + Trace Elements A (2000×, Corning, MT99182CI) + Trace Elements B (2000×, Corning, MT99175CI).

## Immunofluorescence and imaging for cell culture

*Cell culture staining*. Cell cultures were washed twice in PBS and fixed with 4% paraformaldehyde at room temperature for 15 min. Fixed cells were blocked in PBST (PBS containing 0.2% Triton X-100 and 0.5% normal donkey serum (Jackson Immuno Research Laboratories)) at room temperature for 1 h. Primary and

secondary antibodies were diluted in PBST. Cells were incubated with primary antibodies at 4 °C overnight, followed with three rinses, and an incubation with secondary antibodies at room temperature for 1 h. The stained cells were rinsed with PBS and then incubated with DAPI (Sigma-Aldrich) for 2 min to stain the nuclei. Cells were then washed three times by PBS prior to imaging. TUNEL assays for detecting the in situ cell death were performed according to the manufacturer's manual (Roche). For confocal imaging, cells were usually mounted with mounting medium and covered with a cover glass.

**Tissue sectioning and immunohistochemistry**. Cell grafts were fixed with 4% paraformaldehyde at 4 °C for 2 h, followed by three PBS washes at 4 °C (which lasted a few seconds, 10 min and 2 h). The cell grafts were then incubated in 30% (w/vol) sucrose solution at 4 °C overnight. The tissues were embedded in Optimal Cutting Temperature Compound (Tissue-Tek), frozen in liquid nitrogen, and sectioned at 7 μm using a Cryostat (Leica). Section staining was performed by using the same procedure as in "Cell culture staining" without the fixation step.

**Flow cytometry and cell sorting**
*Single-cell suspension from cell cultures*. Differentiated hESC cultures were rinsed with PBS and then incubated with 0.25% trypsin-EDTA (Life Technologies) at 37 °C for 1–3 min. The trypsin was neutralized with mouse embryonic fibroblast medium. The dissociated cells were rinsed twice in PBS or DMEM/F12 medium for further analysis. For intracellular antibody staining, single cells were fixed in 200 μL of BD Cytofix/Cytoperm Buffer (BD Biosciences) at 4 °C for 20 min followed by three washes in BD Perm/Wash Buffer. Fixed cells were incubated in 150 μL of primary antibody buffer at 4 °C for 1 h, followed by 30 min in a secondary antibody (if there is a secondary antibody) buffer after being rinsed twice in Perm/Wash Buffer. Stained cells were washed twice in Perm/Wash Buffer prior to analyses. The acquired data were analyzed by FlowJo. A figure exemplifying the gating strategy was showed in the Supplementary Method.

**Antibody used in this study**. Primary antibodies used for immunofluorescence and immunohistochemistry include the following: NKX6.1 (DSHB, F55A12-c, 1:300), PDX1 (R&D, AF2419, 1:300), INS-APC (Cell Signaling, C27C9, 1:80), NKX6.1-PE (BD, #563023, 1:40), Human C-peptide (Abcam, Ab14181, 1:100), Glucagon (Cell Signaling, 2760 S, 1:400), Glucagon (Abcam, ab82270, 1:100), Insulin (Abcam, ab7842, 1:100), GFP (Abcam, ab6673, 1:300), NEUROD1 (R&D, AF2746, 1:200), and NEUROD1-APC (BD, 563000, 1:50). Secondary antibodies used were as follows: Donkey anti-mouse Alexa Fluor 488 (Jackson Lab, 715-545-151), Donkey anti-goat Alexa Fluor 488 (Jackson Lab, 705-545-147), Donkey anti-rabbit Alexa Fluor 488 (Jackson Lab, 711-545-152), Donkey anti-guinea pig Alexa Fluor 488 (Jackson Lab, 706-545-148), Donkey anti-mouse Alexa Fluor 550 (Invitrogen, SA5-10167), Donkey anti-rat Alexa Fluor 555 (Abcam, ab150154), Donkey anti-rabbit Alexa Fluor Cy3 (Jackson Lab, 711-165-152), Donkey anti-rat Alexa Fluor 647 (Abcam, ab150155), Donkey anti-goat Alexa Fluor 647 (Jackson Lab, 705-605-147), Donkey anti-guinea pig Alexa Fluor 647 (Jackson Lab, 706-605-148), Donkey anti-mouse Alexa Fluor 647 (Jackson Lab, 715-605-151), Donkey anti-guinea pig FITC (Jackson Lab, 706-095-148), Donkey anti-mouse Cy5 (Jackson Lab, 715-175-151), Donkey anti-mouse Alexa Fluor 647 (AF647) (Jackson Lab, 715-605-151), and Donkey anti-rabbit Alexa Fluor 647 (Jackson Lab, 711-605-152). All secondary antibodies were prepared following the manufacturers' instruction and were used as 1:300–500. Image acquisition was performed using a Zeiss LSM 710 confocal microscope. Images were processed using Fiji (ImageJ, v2.0.0) or Zen (Zeiss).

**Gene targeting**. For LoxP-flanked puromycin (Puro) selection, a 2A-NLS-GFP was knocked in and fused in frame to the end of endogenous NKX6.1 coding sequence in H1 hESCs. The Puro cassette was later removed by adding TAT-CRE protein. The cells will show nucleus-localized GFP expression once endogenous NKX6.1 gene is activated. Targeting site of guide RNA for NKX6.1 locus by CRISPR-Cas9: 5′-TGCGGCGGGCGGCGGCGTTC-3′. For targeting, a 1.3 kb left homologous arm and a 2.0 kb right homologous arm were used.

**Static glucose-stimulated insulin secretion**. For static glucose-stimulated insulin secretion assays, stage-8 cells (usually 1–2 clusters, equivalent to ~0.2–0.4 million cells in total), or 15 human islets were rinsed twice with Krebs buffer (129 mM NaCl, 4.8 mM KCl, 2.5 mM CaCl$_2$, 1.2 mM MgSO$_2$, 1 mM Na$_2$HPO$_4$, 1.2 mM KH$_2$PO$_4$, 5 mM NaHCO$_3$, 10 mM HEPES, and 0.1% BSA in deionized water and sterile filtered) and then pre-incubated in Krebs buffer for 60 min. Cells were then incubated in Krebs buffer containing 3.3 mM glucose for 60 min. The cells were then transferred to a new plate containing Krebs buffer with 16.7 mM glucose (or other reagents) for 60 min. Supernatant samples were collected after each incubation period and frozen for ELISA analysis. ELISA kits included human C-peptide ELISA (#10-1141-01; Mercodia).

**Total insulin content measurement**. Human islets, stage-7 and stage-8 cells were incubated in Tris-EDTA (pH 7.4, on ice) and were sonicated to disrupt all cell membranes. After brief centrifugation, an aliquot of the lysed cell suspension (containing hormone) was measured using the insulin ELISA kits (#10-1113-01; Mercodia).

**In vivo animal experiments**. Transplantation assay was conducted as previously described[31]. To be specific, STZ was conducted by intraperitoneal injection (i.p., 35 mg/kg daily) for 4 days, to induce diabetes. Mice were considered to be diabetic when blood glucose measurements were above 250 mg/dl for 4 consecutive days. About 1.6 million cells (equals about 8 clusters, each has ~0.2 million cells) were transplanted under the kidney capsule for each diabetic mouse. "No treatment" was acted as a sham surgery. For in vivo glucose-stimulated C-peptide secretion assay, human C-peptide levels were measured after an overnight fasting and 60 min following an i.p. glucose bolus (2 g glucose/kg body weight (2 g/kg; 30% solution)) at 5 weeks post cell transplantations.

**Statistical analyses**. Before experiments, the sample size was not predetermined using any statistical methods. All values were expressed as the mean ± SEM or mean ± SD as indicated. Statistical analysis was performed using the Prism software (v7 or v8, GraphPad). Graphs were generated using Prism (Graphpad). P-values were determined by t-tests.

**Reporting summary**. Further information on research design is available in the Nature Research Reporting Summary linked to this article.

## Data availability
The raw data associated with this study are provided as a source data file or available from the corresponding author upon further request. Source data are provided with this paper.

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

## Acknowledgements

We thank C. O'Connor of Salk flow cytometry core for cell sorting, and C. Fine and J. Olvera of UCSD HESCCF for technical assistance. We are grateful for U. Manor and Tong Zhang of the Waitt Advanced Biophotonics Core for confocal imaging. We are also thankful to Drs. Jhala, Sander, Wortham, and all the GEL-B lab members for helpful discussion, manuscript preparation, and constructive criticism, and to M. Schwarz and P. Schwarz for administrative assistance. This project was supported by Universidad Católica San Antonio de Murcia (UCAM), Primafrio, The Larry L. Hillblom Foundation, The Moxie Foundation and Diabetes Research Connection (Project Number: 15; Institution Project Number: Liu-DRC-2019).

## Author contributions

H.L. and J.C.I.B. conceived the study, designed experiments, and interpreted results. H.L. and R.L. performed experiments, collected, and analyzed the data. H.L., R.L., H.-K.L., and J.C.I.B. wrote the manuscript H.-K.L., C.W., Y.Y., Z.M., L.S., J.D., A.H., L.M.M., and E.N.D. helped to design the experiments, perform some experiments (e.g., immunofluorescence staining), discuss the results, and write the manuscript. J.C.I.B. supervised the study and wrote the manuscript with comments from all authors.

## Competing interests

The authors declare no competing interests.
