## [Peer Review File · Nature Communications]

REVIEWER COMMENTS

Reviewer #1 (Remarks to the Author):

In this manuscript, the authors describe a new differentiation protocol and report a chemical recipe for increasing the production efficiency of hPSC-derived NKX6.1+/INS+ β -cells. The authors state that their method incorporated several optimizations, including: 1) a V-bottom plate-based aggregation strategy to make 3D pancreatic progenitor clusters; 2) a 10-chemical/factor cocktail (they term "PP-10C") to "poise" pancreatic progenitor clusters; and 3) an additional 3-step chemical combination of signaling modulators to guide pancreatic progenitor clusters toward functional β -cells. Finally, the derived β -cells responded to high glucose stimulation in vitro under static GSIS assay and were able to reduce hyperglycemia in STZ-induced diabetic mice within two weeks post transplantation. Overall, while the high efficiency and functionality of the differentiations is an impressive achievement, the manuscript is lacking important details or data to further substantiate many of the key claims. Therefore, the manuscript could be further strengthened if the following points are addressed.

Major points:

1. Line 88-89 –Results provided do not fully support the conclusion that "The resulting chemically defined protocol produced a monolayer of PPs with an efficiency of ~80% (Figure.1C, S1A, S1B and S1C)". This conclusion appears to be based upon detecting NKX6.1 (or GFP) expression without examining PDX1 expression. The authors should immunostain PDX1 with NKX6.1 and quantify the percentage of PDX1+/NKX6.1+ double positive cells among the whole cell populations at the end of stage 4. Similarly, it is interesting to see the dose of Activin-A at stage 1 impacts NKX6.1 expression at stage 4 (Figure.S1D). It is equally important to examine whether this gradient treatment of Activin-A at stage 1 impacts PDX1 expression, and more importantly, the co-expression of PDX1 and NKX6.1.
2. Line 98-99 – the authors state "we robustly generated compact 3D clusters of PPs with optimal size without significant loss of cells". How do the authors define "optimal size"? The authors added 0.1-0.4 million cells into each well of V-bottom 96-wp according to the seeding conditions described in the methods (Line 293-294). What sizes are the resulting 3D clusters? Based on the images shown in Figure.1E-F, the clusters seem to be ~1 mm in diameter. Were these relatively large clusters examined for dead cells and a necrotic core? Experiments with Live/Dead imaging would help address this. Also regarding Figure.1F, it is hard to appreciate the NKX6.1-GFP fluorescence without a negative control image (e.g. undifferentiated hPSC aggregates), since the image presented is not high resolution enough to distinguish cells/GFP nucleus. A high-resolution image showing NKX6.1-GFP and/or PDX1 expression in PP 3D clusters generated by V-bottom plate aggregation would be appreciated.
3. Figure.1H – can the authors explain why the purple area shown in the top left corner of the "GCG" image does not appear in the "merged with DAPI" image?
4. Line 112: "Consistent with previous reports, most INS+/NKX6.1- cells expressed GCG". The authors ran flow analysis for NKX6.1 and INS, and this should be presented with GCG too.
5. Figure.1J – By screening 21 conditions, the authors found condition #13 best maintained NKX6.1 expression (Line 120-121). However, the PP 3D clusters are relatively large and based on the images shown in Figure 1J, many clusters have empty appearing regions (especially core regions) without NKX6.1 staining. As the authors didn't show DAPI nuclear staining, it is not clear whether cells in the core were still there but didn't maintain NKX6.1 expression or cells were dead/dying in the core and therefore losing NKX6.1. Thus, there is a possibility that condition #13 stands out as the best condition to retain NKX6.1 expression via promoting cell survival within these giant 3D PP clusters.
6. It is appreciated that the authors provided experimental design (Supplementary method 1-2) and corresponding immunostaining data/flow cytometry analysis (Figure.1J, Figure.2B and Figure. S2C) to support their conclusions on optimizing stage 5, stage 6 and stage 8. To further strengthen the protocol developed in this manuscript, it would be more convincing if the authors could provide

screening conditions and rationale for the optimization of stage-7. For example, how does supplementation of HGF, IGFI and PD173074 into stage 7 medium promote stage 6 endocrine progenitors toward commitment to immature β -cells? Since it is emphasized by the authors that they used a "late-stage readout strategy" to optimize the differentiation system, the authors should do retrospective analysis on stage 7 cells by examination of key transcription marker genes.

7. The authors should include the experimental methods for the GSIS study (Figure 2E) in the "Material and Method" to describe how the authors did this static GSIS assay in vitro. For example, how many stage 8 cells were used for one assay and how long were cells incubated the prior to collection of the supernatant for hormone measurement? Aside from the data presented as "fold over basal" shown in Figure 2E, the authors should also show the raw values of C-peptide concentration. Equally importantly, did the authors examine the total insulin or total C-peptide content of stage 8 cells? As positive controls, the authors should examine human C-peptide secretion from primary human islets assessed using the same methods. Dynamic GSIS assay of stage 8 cells with different secretagogue stimulation (e.g. exendin-4 and KCl) would be more convincing evidence to demonstrate their in vitro functionality.

8. Regarding Figure.2F and 2G, how many stage 8 cells and fibroblasts were transplanted under the kidney capsule of STZ-induced diabetic NSG mice? Is the "no treatment" group a sham surgery? What glucose dose were animals given? The authors should include this critical information in the manuscript.

Minor points:

1. Line 33-34 and Line 64 – "The derived β -cells were physiologically functional" is not accurate, since the GSIS study (Figure 2E) adopted a supra-physiological concentration of glucose challenge (16.7 mM) as the stimulation.
2. Line 126-127 – "Further characterization of PP-10C-treated PPs proved that most NKX6.1+ cells expressed PDX1". In addition to the flow cytometry data shown in Figure.1M, it would be further strengthened if the authors could show immunostaining data of NKX6.1 with PDX1 to draw that conclusion.
3. Line 271-272 – the authors used "3ug/ml chir99021" for stage 1 day 1 and "0.3ug/ml chir99021" for stage 1 day 2. By converting into uM, these are 6uM CHIR99021 and 0.6uM CHIR99021. Compared with many other published protocols, 3uM CHIR99021 is the most commonly used concentration at stage 1. On the other hand, it seems the authors only tested 1uM and 3uM CHIR99021 as listed in the Supplementary Table 1. Can the authors explain the rationale of deciding to use a higher concentration of CHIR99021 (6uM)?
4. Line 281-284 – the authors missed providing the concentration of KGF used in the stage 3 recipe.
5. Line 297 and Line 322 – do the authors really use "ZnS04 (10mM)" in the differentiation protocol, or should it be 10uM?

Reviewer #2 (Remarks to the Author):

The reporter cell line is on the H1 hESC background. H1 has a high propensity towards generating pancreatic lineage among hPSC lines. The authors show flow cytometry plots of two hPSC lines differentiated towards INS/NKX6/1+ cells using their approach. How many hPSC lines were tested together? What was the heterogeneity between lines? Importantly, how similar were independent differentiation batches for the same hPSC line and among different tested lines.

The authors hypothesize that premature NKX6.1 loss causes the lower efficiency of beta cell induction with other protocols. Another likely scenario is that NKX6.1 expression is induced only in a portion of PDX1+ PPs cells, especially since PDX1 expression is very efficiently induced (even in ~ 90% of total

live cells) at this stage of differentiation. Therefore, it is essential to show which experimental data, like lineage tracing or by other methods, support this claim. Further, at what stage is the expression of NKX6.1 decreasing?

How reproducible is the described protocol? The data showing INS/NKX6.1 induction from several rounds of independent differentiation preferable for more than H1 cell line, would be critical as variability is a common problem.

Have the authors assessed the long-term efficacy and safety of stage 8 grafts? What was the longest time in vivo tested?

Fig 1: K and L- NGN3 and NEUROD1 images not possible to see any positively stained cells. NEUROD1 staining in Suppl. figure is more convincing. Please replace them with other images or remove the claim.

Fig 2. E- GSIS in vivo- the data cannot be presented as fold changed, but amounts of secreted c-peptide should be shown along with positive control: human primary islets. Otherwise, these critical data are almost meaningless.

Fig 2F and G: The critical positive control human cadaveric islets and possible cells derived with other published protocols are missing. The most recent protocols are from Melton, Hebrok, or Millman.

Reviewer #3 (Remarks to the Author):

Review: In this manuscript by Liu et al the authors report that a new protocol used to develop organoids with 10 chemicals leads to a greater efficiency of formation of beta-like cells compared to the existing approaches. They report that the beta-like cells are functionally competent and able to reverse hyperglycemia in a model of STZ-induced diabetes. The data are interesting but lack a number of experimental and technical details and don't provide a mechanistic explanation for the intriguing findings.

Critique:

1. Introduction – last line should be rewritten.
2. What was the rationale for using the custom library of compounds?
3. Under Results section (Figure 2) the authors should spell out in mM the exact glucose concentration when they mention "low" and "high". Makes it easier for the reader rather than having to go back forth between text and figure to see the values. Figure 2E actually shows C-peptide. Please show insulin. Please include absolute levels of C-peptide and insulin so the readers can appreciate the levels. How does this value compare to C-peptide secretion from native islets at this glucose concentration. It would be useful to plot C-peptide and insulin secretion data in the same bar chart so one can compare differential secretory response between the organoid-derived b-cells and the native islet beta cells to glucose. The glucose level of 16.7 mM is pretty high. While this concentration is routinely used in the community it will be useful to know whether these cells respond to 7-8 mM glucose (i.e. post-prandial levels)? Now, that would be a very important advance in the field!
4. The authors have focused only on glucose as a secretagogue. While glucose is one of the most important what about responses to GLP-1 stimulation?
5. Considering there are double hormone+ cells, did the authors examine glucagon levels in the secretory functional responses? These cells have been reported to occur in most protocols. Did these cells functionally respond in this organoid approach?
6. The Nkx6.1/GFP merged cells in Fig 1C does not appear uniform in contrast to Fig 1B. Is there an explanation?

7. In Figure 1I the % cells don't add up to 100%!
8. In Figure 1J there are large empty spaces in the middle in the clusters in #s 1, 4, 15 and 19. Are these artefacts?
9. In Figure 2 F, a control model with native islets being transplanted would provide a useful comparison to assess how the organoid-derived beta-like cells act to counter the hyperglycemia. This is especially important since the mouse recovery from the stress after the surgery typically takes 10 days. The fact that the blood glucose comes down to less than 250 mg/dl within a week after transplanting the organoid-derived beta-like cells is a remarkable observation.
10. Not sure if this Reviewer missed it - how many beta-like cells were transplanted? How does this compare with the typical transplantation of human IEQs in a similar STZ model setting?
11. While the data are intriguing no mechanistic explanations are provided as to how the organoid approach is able to promote the efficiency and/or the secretory function of the beta-like cells.

REVIEWER COMMENTS

Reviewer #1 (Remarks to the Author):

In this manuscript, the authors describe a new differentiation protocol and report a chemical recipe for increasing the production efficiency of hPSC-derived NKX6.1+/INS+ β -cells. The authors state that their method incorporated several optimizations, including: 1) a V-bottom plate-based aggregation strategy to make 3D pancreatic progenitor clusters; 2) a 10-chemical/factor cocktail (they term “PP-10C”) to “poise” pancreatic progenitor clusters; and 3) an additional 3-step chemical combination of signaling modulators to guide pancreatic progenitor clusters toward functional β -cells. Finally, the derived β -cells responded to high glucose stimulation in vitro under static GSIS assay and were able to reduce hyperglycemia in STZ-induced diabetic mice within two weeks post transplantation. Overall, while the high efficiency and functionality of the differentiations is an impressive achievement, the manuscript is lacking important details or data to further substantiate many of the key claims. Therefore, the manuscript could be further strengthened if the following points are addressed.

Major points:

1. Line 88-89 –Results provided do not fully support the conclusion that “The resulting chemically defined protocol produced a monolayer of PPs with an efficiency of ~80% (Figure.1C, S1A, S1B and S1C)”. This conclusion appears to be based upon detecting NKX6.1 (or GFP) expression without examining PDX1 expression. The authors should immunostain PDX1 with NKX6.1 and quantify the percentage of PDX1+/NKX6.1+ double positive cells among the whole cell populations at the end of stage 4.

Response: We followed the reviewer’s suggestion to perform the immunostaining of PDX1 with NKX6.1 in sections from the aggregate at the end of stage 4, which is shown in the new Figure S1D. In addition, we performed FACS analysis to quantify the percentage of PDX1+/NKX6.1+ double positive cells at the end of stage 4 (see newly added data in Figure S1F). The results of three biological replications showed an average of ~80% PDX1+/NKX6.1+ double positive cells among the whole cell populations at the end of stage 4. These data collectively lend additional support to our conclusion that “The resulting chemically defined protocol produced a monolayer of PPs with an efficiency of ~80%”.

Similarly, it is interesting to see the dose of Activin-A at stage 1 impacts NKX6.1 expression at stage 4 (Figure.S1D). It is equally important to examine whether this gradient treatment of Activin-A at stage 1 impacts PDX1 expression, and more importantly, the co-expression of PDX1 and NKX6.1.

Response: We performed the additional analysis and included these data in the new Supplementary Table 1. When the decreasing gradient (day1-day2-day3: 115-110-110 ng/ml) dose of Activin-A was administered at stage 1, of all NKX6.1-expressing cells, more than 98.5% of them also express PDX1 at the end of stage 4. This suggests that NKX6.1-expressing cells usually are a subset of PDX1-expressing cells. When constant low-dose or high-dose of Activin-A was used at stage-1, both the percentage of PDX1-expressing cells and NKX6.1-expressing cells at stage-4 were affected. Below is a summary of how the doses of Activin-A affect PDX1 and/or NKX6.1 expressions.

Activin-A (ng/ml) day1-day2-day3	PDX1-positive cells	NKX6.1- positive cells	PDX1-positive/ NKX6.1-positive
------------------------	---------------------------	-----------------------------------

			cells
Constant low-dose of Activin-A (100-100-100)	~50%-80%	~15%-50%	~15%-50%
Decreasing gradient dose of Activin-A (115-110-100)	~93%	~80%	~80%
Constant high-dose of Activin-A (115-115-115)	~90%	~20%-50%	~20%-50%

Based on these observations, we speculate that the constant low-dose Activin-A does not efficiently induce hESCs into definitive endoderm at stage-1, thus generating a lot of non-pancreatic cells. The constant high-dose Activin-A successfully induced hESCs into definitive endoderm at stage-1 and generated a high percentage of PDX1-expressing cells. However, high-dose Activin-A severely reduced cell proliferation and resulted in a suboptimal cell density, both of which impacted NKX6.1 gene expression at stage-4. We frequently observed that NKX6.1+ cells tended to appear in the high-density area of stage-4 cell cultures, whereas the low cell density at the end of stage-1 tended to lead to a low percentage of NKX6.1+PDX1+ cells at the end of stage-4. We had added these descriptions into the updated manuscript as new Supplementary Table 1.

2. Line 98-99 – the authors state “we robustly generated compact 3D clusters of PPs with optimal size without significant loss of cells”. How do the authors define “optimal size”? The authors added 0.1-0.4 million cells into each well of V-bottom 96-wp according to the seeding conditions described in the methods (Line 293-294). What sizes are the resulting 3D clusters? Based on the images shown in Figure.1E-F, the clusters seem to be ~1 mm in diameter. Were these relatively large clusters examined for dead cells and a necrotic core? Experiments with Live/Dead imaging would help address this.

Response: We agree with the reviewer that “optimal size” is not a clear term in this context. We determined the number of cells and the size of the 3D PP clusters based on the following factors: (1) the cells should form a compact cluster without significant loss of cells after aggregation; and (2) the clusters are in the size that can be easily handled manually by an opened 100ul pipette tip or tweezers in the medium for transferring onto new air-liquid interfaces when beginning the next differentiation stages. For this, we tested a different number of seeding cells and found about 0.1-0.4 million cells, which generated clusters with a diameter ~0.5~3mm and fulfilled the above-mentioned criteria. If more than 0.5 million cells were used, significant cells did not completely integrate into the clusters during the aggregation process. If less than 0.1 million cells were used for making clusters, the clusters were too small for later stage manual handling. Typically, we used 0.2 million cells for making one cluster. It’s noteworthy that these pp-3D-clusters will become a little flat (like a very thick disk) after further culturing in the air-liquid interface.

We also followed the reviewer’s suggestion of testing cell survival in the clusters. We examined these clusters after culturing for 4 days in different conditions. To identify cell death by live/dead imaging, we applied the TUNEL assay and co-stained with NKX6.1 and DAPI. The results revealed very few apoptotic cells in PP-10C conditions (see newly added data in Figure S1H, condition #13). In contrast, we did observe more cell deaths in other conditions. Moreover, we found necrotic cores (with a lot of pervading DAPI, dying/dead cells and cell debris) in some other conditions, such as conditions #4, #5,

and #19 in Figure S1H. Very occasionally, we saw a small cyst inside PP-10C treated clusters (condition #13) (No cyst in Figure 1J or 1K; a small cyst in Figure S1H). As the small cyst in condition #13 has sharp boundaries and contains very few dead or dying cells, or cell debris/fragments, we believe these cysts might be early-stage pancreatic duct-like structures instead of necrotic cores.

Also regarding Figure.1F, it is hard to appreciate the NKX6.1-GFP fluorescence without a negative control image (e.g. undifferentiated hPSC aggregates), since the image presented is not high resolution enough to distinguish cells/GFP nucleus. A high-resolution image showing NKX6.1-GFP and/or PDX1 expression in PP 3D clusters generated by V-bottom plate aggregation would be appreciated.

Response: According to the reviewer's suggestion, we added undifferentiated NKX6.1-NLS-GFP hPSCs aggregations as a negative control (newly added data as Figure 1F). In addition, we also generated a high-resolution image showing PDX1-expression in PP-3D-clusters (Figure S1I).

3. *Figure.1H – can the authors explain why the purple area shown in the top left corner of the “GCG” image does not appear in the “merged with DAPI” image?*

Response: We have corrected this mistake in the revision.

4. *Line 112: “Consistent with previous reports, most INS+/NKX6.1- cells expressed GCG”. The authors ran flow analysis for NKX6.1 and INS, and this should be presented with GCG too.*

Response: According to the reviewer's suggestion, we have newly added the result of INS/GCG flow cytometry analysis from the end-stage of the Method-1 differentiation cells. Together with the data provided previously (Figure 1H and 1I), this clearly supports that “most INS+/NKX6.1- cells expressed GCG”.

5. *Figure.1J – By screening 21 conditions, the authors found condition #13 best maintained NKX6.1 expression (Line 120-121). However, the PP 3D clusters are relatively large and based on the images shown in Figure 1J, many clusters have empty appearing regions (especially core regions) without NKX6.1 staining. As the authors didn't show DAPI nuclear staining, it is not clear whether cells in the core were still there but didn't maintain NKX6.1 expression or cells were dead/dying in the core and therefore losing NKX6.1. Thus, there is a possibility that condition #13 stands out as the best condition to retain NKX6.1 expression via promoting cell survival within these giant 3D PP clusters.*

Response: To address this question, we performed NKX6.1/TUNEL/DAPI staining for ten conditions (including #13 and the other nine ones). The TUNEL staining showed very few dying/dead cells in 10-C conditions (#13 condition) (see newly added data in Figure S1H), but we did observe more dying/dead cells in most of other conditions.

Moreover, in some other conditions (Figure S1H, condition #4, #5, and #19) we found necrotic cores (with lots of pervading DAPI stain, dying/dead cells, and cell debris). As mentioned in the previous answer, in some occasional cases, we saw a small cyst inside PP-10C treated clusters in the condition #13 (No cyst in Figure 1J or 1K; a small cyst in Figure S1H). As the small cyst in the condition #13 has sharp boundaries and contains very few dead or dying cells, or cell debris/fragments, we believe these cysts might be pancreatic duct-like structures instead of necrotic cores.

In light of these observations, we agree with the reviewer that condition #13 could retain NKX6.1 expression via promoting cell survival. But we also observed that in conditions #1 and #15 (Figure S1H), a lot of live cells lost their expression of NKX6.1 despite the fact that cells survived well and the aggregates had no obvious necrotic core. Thus, we believe that besides promoting cell survival, condition #13 also employs some other mechanisms to retain NKX6.1-expression.

6. It is appreciated that the authors provided experimental design (Supplementary method 1-2) and corresponding immunostaining data/flow cytometry analysis (Figure.1J, Figure.2B and Figure. S2C) to support their conclusions on optimizing stage 5, stage 6 and stage 8. To further strengthen the protocol developed in this manuscript, it would be more convincing if the authors could provide screening conditions and rationale for the optimization of stage-7. For example, how does supplementation of HGF, IGF1 and PD173074 into stage 7 medium promote stage 6 endocrine progenitors toward commitment to immature β -cells? Since it is emphasized by the authors that they used a “late-stage readout strategy” to optimize the differentiation system, the authors should do retrospective analysis on stage 7 cells by examination of key transcription marker genes.

Response: We are glad to hear that the reviewer appreciated our efforts in optimizing the different stages of the differentiation processes. For stage-7, we applied different treatments on stage-7 cells, and then subject cells for stage-8 differentiation. We analyzed NKX6.1+/INS+ cells percentage at the end of stage 7 and 8 and performed GSIS at the end of stage-8. The key chemical combinations we tested at stage 7 were $i\beta$ -9C minus LDN, T3, Repsox, GSIXX, RA, HGF, IGF1, or PD, respectively. We found that LDN, T3, Repsox, GSIXX, RA, HGF, IGF1, and PD is all required for maintaining a high percentage of INS+ and NKX6.1+ cells at the end of stage 8, and the removal of some of them also affects the GSIS function of the final product- β cells. The addition of $i\beta$ -9C is used for the first time at stage-7 but not in other published protocols. However, compared to our analysis of other stages, the screening experiments for stage-7 were not systematic but instead were done in multiple small experiments. Therefore, we found it difficult to summarize them into a single table. Although we agree that analyzing how these $i\beta$ -9C factors affect cells’ transcriptional factors at stage 7 is interesting. Such a detailed analysis will be better suited to a follow-up study.

7. The authors should include the experimental methods for the GSIS study (Figure 2E) in the “Material and Method” to describe how the authors did this static GSIS assay in vitro. For example, how many stage 8 cells were used for one assay and how long were cells incubated the prior to collection of the supernatant for hormone measurement?

Response: We have added the following method details in the updated manuscript. For static glucose-stimulated insulin secretion assays, Stage-8 cells (usually 1-2 clusters, equivalent to 0.2~0.4 million cells in total) (Figure 3A), or 15 primary human islets (Figure 3A) were rinsed twice with Krebs buffer (129mM NaCl, 4.8mM KCl, 2.5 mM CaCl₂, 1.2 mM MgSO₂, 1mM Na₂HPO₄, 1.2mM KH₂PO₄, 5mM NaHCO₃, 10mM HEPES, and 0.1% BSA) and then pre-incubated in Krebs buffer for 60 mins. Cells were then incubated in Krebs buffer containing 3.3 mM glucose for 60 mins. The cells were then transferred to a new plate containing Krebs buffer with 16.7mM glucose (or other reagents) for an additional 60 mins. Supernatant samples were collected after each incubation period and frozen prior to ELISA analysis. ELISA kits including human C-peptide ELISA (#10-1141-01; Merckodia).

Aside from the data presented as “fold over basal” shown in Figure 2E, the authors should also show the raw values of C-peptide concentration.

Response: According to the comments from this reviewer and other reviewers, we presented the absolute values of C-peptide concentrations in the revised figures (Figure 3A). These values are normalized to account for cell number differences between tests.

Equally importantly, did the authors examine the total insulin or total C-peptide content of stage 8 cells?

Response: According to the reviewer’s suggestion, we examined the total insulin content of stage 8 cells (Figure S3B), which is ~ 62ng per 10000 cells. This value is comparable to that of human primary islets.

As positive controls, the authors should examine human C-peptide secretion from primary human islets assessed using the same methods. Dynamic GSIS assay of stage 8 cells with different secretagogue stimulation (e.g. exendin-4 and KCl) would be more convincing evidence to demonstrate their in vitro functionality.

Response: According to the reviewers’ suggestion and to better demonstrate the functionalities of β cells from our new protocol, we added primary human islet as a positive control and performed additional secretagogue stimulations including Exendin-4 and KCl in our stage-8 cells. This new data showed that β cells from our new protocol performed very similarly to primary islet cells in static GSIS assay (Figure 3A and S3A). For Dynamic GSIS assay, we had tried but were not able to find suitable equipment in the surrounding areas. Although dynamic GSIS assay could provide a little more information for assessing the function of our cells, our static GSIS assay data should be sufficient to support the conclusion that the beta cells from our new protocol are functional.

8. Regarding Figure.2F and 2G, how many stage 8 cells and fibroblasts were transplanted under the kidney capsule of STZ-induced diabetic NSG mice? Is the “no treatment” group a sham surgery? What glucose dose were animals given? The authors should include this critical information in the manuscript.

Response: We routinely transplanted about 1.6 million cells (about 8 clusters) under the kidney capsule for each diabetic mouse. “No treatment” is a sham surgery. For the *in vivo* glucose-stimulated c-peptide secretion assay, 2g glucose/kg body weight (2g/kg; 30% solution) was used after 16hr fasting. This information has been added to the new manuscript.

Minor points:

1. Line 33-34 and Line 64 – “The derived β -cells were physiologically functional” is not accurate, since the GSIS study (Figure 2E) adopted a supra-physiological concentration of glucose challenge (16.7 mM) as the stimulation.

Response: We have removed the word “physiologically” from the manuscript.

2. Line 126-127 – “Further characterization of PP-10C-treated PPs proved that most NKX6.1+ cells expressed PDX1”. In addition to the flow cytometry data shown in Figure.1M, it would be further

strengthened if the authors could show immunostaining data of NKX6.1 with PDX1 to draw that conclusion.

Response: We have added this new data in Figure S11.

3. Line 271-272 – the authors used “3ug/ml chir99021” for stage 1 day 1 and “0.3ug/ml chir99021” for stage 1 day 2. By converting into uM, these are 6uM CHIR99021 and 0.6uM CHIR99021. Compared with many other published protocols, 3uM CHIR99021 is the most commonly used concentration at stage 1. On the other hand, it seems the authors only tested 1uM and 3uM CHIR99021 as listed in the Supplementary Table 1. Can the authors explain the rationale of deciding to use a higher concentration of CHIR99021 (6uM)?

Response: We are sorry for these typos. The CHIR99021 concentration should be “3 μM” for day 1 and “0.3 μM” for day 2. We have corrected these typos in the updated manuscript. We thank the reviewer for pointing these out.

4. Line 281-284 – the authors missed providing the concentration of KGF used in the stage 3 recipe.

Response: We have added this information (KGF, 50ng/ml) in the updated manuscript.

5. Line 297 and Line 322 – do the authors really use “ZnS04 (10mM)” in the differentiation protocol, or should it be 10uM?

Response: We are sorry for this typo. It should be “10 μM”. We have corrected this in the updated manuscript. We thank the reviewer for pointing this out.

Reviewer #2 (Remarks to the Author):

The reporter cell line is on the H1 hESC background. H1 has a high propensity towards generating pancreatic lineage among hPSC lines. The authors show flow cytometry plots of two hiPSC lines differentiated towards INS/NKX6/1+ cells using their approach. How many hPSC lines were tested together? What was the heterogeneity between lines? Importantly, how similar were independent differentiation batches for the same hPSC line and among different tested lines;

Response: We have tested in total five cell lines (including H1, H1-NKX6.1-GFP; and three integration-free hiPSC lines (hiPSC1, hiPSC2, and hiPSC3) for our protocol. We were able to routinely generate >60% beta cells from them. Below is a summary table of the tests done in these five cell lines. In addition, our protocol is reproducible for the same cell line with low batch variations. These results suggest our protocol would be suitable for many ES and high-quality iPS cell lines. These data have been added to the new manuscript as Supplementary Table 3.

Cell line	Efficiency of β cell generations in different tests
-----------	---

H1 hESC	78%; 82%; 67%; 75%; 74%
H1 hESC-NKX6.1-GFP	75%; 64%; 77%
hiPSC1	61%; 57%
hiPSC2	68%; 60%; 63%
hiPSC3	63%

The authors hypothesize that premature NKX6.1 loss causes the lower efficiency of beta cell induction with other protocols. Another likely scenario is that NKX6.1 expression is induced only in a portion of PDX1+ PPs cells, especially since PDX1 expression is very efficiently induced (even in ~ 90% of total live cells) at this stage of differentiation. Therefore, it is essential to show which experimental data, like lineage tracing or by other methods, support this claim. Further, at what stage is the expression of NKX6.1 decreasing?

Response: There might be some misunderstanding here. For all testing after stage 4, we started with a high percentage of a Pdx1+/NKX6.1+ (~80%) (Figure 1C, 1G, S1C, and S1D. We then subjected these high quality PPs (~80% NKX6.1+ cells) to the last three steps of the R-protocol (we termed this combined protocol as “Method 1”). However, this trial yielded only 14% NKX6.1+/INS+ cells (figure 1I). Moreover, the total “NKX6.1+ cells” drop to 19% (Figure 1I). Based on this observation, we speculated that the last three steps of the R-protocol cannot maintain NKX6.1 expression and primarily led to the INS+/GCG+/NKX6.1- cell fate. Importantly, when PP 3D clusters were firstly incubated with PP-10C condition for four days before being subjected to the final three steps of the R-protocol, NKX6.1 expression in the whole cell population was preserved. This suggested that NKX6.1 expression is not stable after induction at the PP stage and can be lost prematurely during the early phase of later differentiation. We have clarified this in the new manuscript. Using our optimized protocol, we did not see a significant decrease of NKX6.1 expression after it was turned on. However, its expression could decrease in a few days if the condition is not ideal, such as many examined conditions in Figure 1J and S1H.

How reproducible is the described protocol? The data showing INS/NKX6.1 induction from several rounds of independent differentiation preferable for more than H1 cell line, would be critical as variability is a common problem.

Response: As we showed in the above table, we have tested in five cell lines (including H1, H1-NKX6.1-GFP, hiPSC1, hiPSC2, and hiPSC3) and multiple rounds for our protocol. We can routinely generate >60% beta cells in those cases.

Have the authors assessed the long-term efficacy and safety of stage 8 grafts? What was the longest time in vivo tested?

Response: The longest time we tested *in vivo* was 8 months after grafting of stage 8 cells. The ES-derived beta cells could reverse diabetes and maintain normal glucose levels for at least 8 months, and did not form tumors.

Fig 1: K and L- NGN3 and NEUROD1 images not possible to see any positively stained cells. NEUROD1 staining in Suppl. figure is more convincing. Please replace them with other images or remove the claim.

Response: It is normal to see few (if not absolutely none) “NGN3” or “NEUROD1” in Figure 1k and 1L, as they should not be efficiently induced under condition #13 at that stage. Figure S2D has a lot of “NEUROD1” is because it was intentionally induced. We revised the descriptions to make it more clear.

Fig 2. E- GSIS in vivo- the data cannot be presented as fold changed, but amounts of secreted c-peptide should be shown along with positive control: human primary islets. Otherwise, these critical data are almost meaningless.

Response: First of all, the former Fig2.E was data from *in vitro* GSIS assays, not *in vivo*. Following the comments from reviewers, we presented the absolute values of C-peptide concentrations in the revised figures (new Figure 3A). These values are normalized to account for cell number differences between tests. In addition, following the reviewers’ suggestions and to better demonstrate the functionalities of beta cells from our new protocol, we added primary human islets as positive controls and performed additional secretagogue stimulations, including Exendin-4 and KCl in our stage-8 cells (Figure S3A). Collectively, this data showed that beta cells from our new protocol performed very similarly in static GSIS assays as primary islet cells (Figure 3A and S3A). Moreover, we also examined the total insulin content of stage 8 cells with controls from primary human islets (Figure S3B)

Fig 2F and G: The critical positive control human cadaveric islets and possible cells derived with other published protocols are missing. The most recent protocols are from Melton, Hebrok, or Millman.

Response: The recent protocols from Melton, Hebrok, or Millman all have made some breakthroughs and addressed issues in their interests. In this study, we focused more on the special issues to achieve high efficiency, homogeneity, and cell line independency. That was why we strived to develop our own protocol. It will be not very feasible to compare cells differentiating from all different protocols for *in vivo* transplantation experiments, considering that all those protocols involve multiple steps and several dozen chemicals/factors and require high amounts of differentiated cells for *in vivo* assays. Therefore, it would be extremely time-consuming and cost prohibitive to repeat those protocols merely for the purposes of making them as control. We think such comparisons will be better pursued through collaborations in the future. Obviously, this is beyond the scope of our study at the moment. To acknowledge their contributions to the field, we have cited these recent publications in our revised manuscript as collective efforts to advance the field.

We agree with the reviewer that including a human islet positive control is important. Due to the current COVID19 pandemic, we are facing a lot of difficulty in securing a sufficient number of samples as well as lab hours. Specifically, we rely on external providers for human islet samples. As far as we know, the main supplier of US human islet distribution was completely shut down for several months since the start of pandemic outbreak, and has recently resumed operations but with a much reduced capacity. In addition, our Institute is still operating on a reduced capacity, especially in regards to the working hours in animal facilities. Despite these difficulties, we did our best to obtain human islets and conduct the *in vitro* GSIS assays and total insulin content for human islet controls, as we show in the new figure 3A and 3B.

Nonetheless, COVID19 should not be an excuse to lower the bar. We have some thoughts on the necessity of performing the *in vivo* experiments with human cadaver islets and would like to share them with you here. After going over the literature and communicating with several experts in the field, we have the impression that the results of such transplantation assays of human islets have been accepted as a scientific fact in the field. Consistent with this, we noted that several important publications on hESC-differentiated beta cells recently published in high profile journals, including Vegas, A. et al., Nature Medicine, 2016; Ghazizadeh et al., Nature Communication, 2017; Rosado-Olivieri et al., Nature Communication, 2019, did not provide primary human islets as such *in vivo* controls, and mainly presented the critical comparison between transplantation of hPSC-derived beta cells and no treatments (in addition to that, we also included a more rigorous control of transplantation of fibroblasts). We think that this practice will become more popular in the coming future as it saves time and cost.

Reviewer #3 (Remarks to the Author):

Review: In this manuscript by Liu et al the authors report that a new protocol used to develop organoids with 10 chemicals leads to a greater efficiency of formation of beta-like cells compared to the exiting approaches. They report that the beta-like cells are functionally competent and able to reverse hyperglycemia in a model of STZ-induced diabetes. The data are interesting but lack a number of experimental and technical details and don't provide a mechanistic explanation for the intriguing findings.

Critique:

1. Introduction – last line should be rewritten.

Response: This sentence had grammar issues, and we have rewritten the sentence to “The resulting β cells were functional and capable of reversing hyperglycemia of diabetic model mice within two weeks.”

2. What was the rational for using the custom library of compounds?

Response: Inducing β cells from pluripotent stem cells are very complicated and expensive experiments, which involves multiple stages and chemicals/ growth factors for each stage, thus making a very large-scale screen not feasible. While small commercial chemical libraries that contain hundreds of chemicals are also available, they usually fail to contain chemicals/growth factors regulating all classic developmental pathways. To reduce cost and save time, we generated a custom screen library comprised of more than one hundred chemicals/growth factors that could modulate (activate or inhibit) most of the known development and differentiation-related signaling pathways (former Supplementary Table 1, now Supplementary Table 2). To this end, we managed to screen > 2000 conditions using different combinations of chemicals/factors from the library to finally build up our own protocol.

3. Under Results section (Figure 2) the authors should spell out in mM the exact glucose concentration when they mention “low” and “high”. Makes it easier for the reader rather than having to go back forth between text and figure to see the values. Figure 2E actually shows C-peptide. Please show insulin. Please include absolute levels of C-peptide and insulin so the readers can appreciate the levels. How does this value compare to C-peptide secretion from native islets at this glucose concentration. It would

be useful to plot C-peptide and insulin secretion data in the same bar chart so one can compare differential secretory response between the organoid-derived b-cells and the native islet beta cells to glucose. The glucose level of 16.7 mM is pretty high. While this concentration is routinely used in the community it will be useful to know whether these cells respond to 7-8 mM glucose (i.e. post-prandial levels)? Now, that would be a very important advance in the field!

Response: In the revised manuscript, we have substituted the overly-general terms “low” or “high” glucose concentration with specific numeric descriptions. For the former Figure 2E, although we call it glucose stimulated insulin secretion (GSIS) assay; we actually measure the secreted form of insulin, which is c-peptide. To alleviate this misunderstanding, we changed the assay name “GSIS” to “glucose stimulated C-peptide secretion” assay. As other reviewers also have the same concern, we now use absolute levels of C-peptide for previous Figure 2E (Now Figure 3A). Additionally, we also added primary human pancreatic islets as the control in these assays. These values of our stage 8 cells are comparable to that of native human islets (see the new Figure 3A). As 16.7 mM glucose is most commonly used in the research community, we stuck with this concentration. We did stimulate our stage 8 cells using 8mM glucose for one of the prior experiments and observed C-peptide induction but a little lower level than that with 16.7mM glucose.

4. The authors have focused only on glucose as a secretagogue. While glucose is one of the most important what about responses to GLP-1 stimulation?

Response: We did not try GLP-1 stimulation, but instead we have tried Extendin-4, a very potent GLP-1 receptor agonist. We showed that our stage 8 cells also responded to Extendin-4 (new Fig S3A), so I would expect that our stage 8 cells would also respond to GLP-1.

5. Considering there are double hormone+ cells, did the authors examine glucagon levels in the secretory functional responses? These cells have been reported to occur in most protocols. Did these cells functionally respond in this organoid approach?

Response: The double hormone+ cells exist in most published protocols with varying percentages. A previous study had been specifically dedicated to investigating such cell population (Bruin, J.E., et al., Stem Cell Research, 2014. 12(1): p. 194-208). They have shown, in agreement with other published papers and our experiences, these double hormone+ cells are not physiologically functional cells. They are neither functional alpha cells nor beta cells. By using our complete protocol, the yield of such cells is very limited. As shown in newly added flow data (Figure S2E), only ~5% double hormone+ cells existed at the end of Stage 8. Therefore, we did not further examine the function of these double hormone+ cells rigorously. Based on our previously experiences, as long as the cells still keep a hybrid phenotype (expressing both GCG and INS), we do not think that our approach or other approaches will make these double hormone+ cells functional.

6. The Nkx6.1/GFP merged cells in Fig 1C does not appear uniform in contrast to Fig 1B. Is there an explanation?

Response: Due to the low brightness of DAPI staining in the previous Figure 1B, the total cell number of the sample was not clear and therefore it could have given the wrong impression that the NKX6.1

expression is uniform across the sample. We have increased the intensity of the DAPI signal in the revised Figure 1B. It is now clear that quite a number of cells were not expressing NKX6.1. Compared to the NKX6.1 expression in Figure 1C, Figure 1B has a lower percentage of NKX6.1+ cells and appears less uniform.

7. In Figure 1I the % cells don't add up to 100%!

Response: We have corrected this mistake in the figure.

8. In Figure 1J there are large empty spaces in the middle in the clusters in #s 1, 4, 15 and 19. Are these artefacts?

Response: We are sorry for the confusion. For these empty spaces in the middle of the clusters in Figure 1J, most of them were filled with cells, while only a few of them have empty spaces or dead cells. We have added several images (including samples from #1, 4, 15 and 19) with DAPI and TUNEL staining to show their conditions in the revised Figure S1H.

9. In Figure 2 F, a control model with native islets being transplanted would provide a useful comparison to assess how the organoid-derived beta-like cells act to counter the hyperglycemia. This is especially important since the mouse recovery from the stress after the surgery typically takes 10 days. The fact that the blood glucose comes down to less than 250 mg/dl within a week after transplanting the organoid-derived beta-like cells is a remarkable observation.

Response: We agree that including a human islet positive control is informative. However, due to the current COVID19 pandemic, we are facing a lot of difficulty in securing a sufficient number of samples as well as lab hours. Specifically, we rely on external providers for human islet samples. As far as we know, the main supplier of US human islet distribution was completely shut down for several months since the outbreak of the pandemic and has recently resumed operations but with much reduced capacity. In addition, our Institute is still operating on a reduced capacity. Despite these difficulties, we did our best to obtain the human islets and conduct the *in vitro* GSIS assay for the human islet control and present this data in the new figure 3A and 3B.

To get around this hurdle, we had contacted several groups in the diabetes field to seek their collaboration for such an *in vivo* experiment. However, it turned out to be extremely difficult to convince anyone to take on such an expensive and labor-intensive experiment with current COVID19 restrictions. Therefore, we think it is not very likely that we can perform the *in vivo* experiments in the foreseeable future. Nonetheless, COVID19 should not be an excuse to lower the bar. We have some thoughts on the necessity of performing the *in vivo* experiments and would like to share them with you here. After going over the literature and communicating with some experts in the field, we have a common thought that the results of such transplantation assays of human islets have been accepted as a scientific fact in the field. Consistent with this, we noted that several important publications on hESC-differentiated beta cells published in high profile journals, including Vegas, A. et al., *Nature Medicine*, 2016; Ghazizadeh et al., *Nature Communication*, 2017; Rosado-Olivieri et al., *Nature Communication*, 2019, did not provide primary human islets as such *in vivo* control, and just presented the critical comparison between transplantation of hPSC-derived beta cells and no treatments (in addition we also

included a more rigorous control of transplantation of fibroblasts). We think this practice will become more reasonable and popular in the coming future as it saves overall cost and time.

10. Not sure if this Reviewer missed it - how many beta-like cells were transplanted? How does this compare with the typical transplantation of human IEQs in a similar STZ model setting;

Response: About 1.6 million cells (about 8 clusters) were transplanted under the kidney capsule of each diabetic mouse. For the transplantation of human islets in a similar STZ model setting, typically 500-4000 IEQs were used. On average, each islet contains about 1560 cells; 500-4000 IEQs have 0.8-6 million cells. From the literature and our past experiences, we feel more human islet cells (rather than ES-derived beta cells) are needed to achieve similar *in vivo* function partially because primary islet cells do not survive very well after transplantation.

11. While the data are intriguing no mechanistic explanations are provided as to how the organoid approach is able to promote the efficiency and/or the secretory function of the beta-like cells.

Response: We are also very interested in the mechanistic explanations but feel such a study is beyond the scope of the current work and better suited for a follow-up.

REVIEWER COMMENTS

Reviewer #1 (Remarks to the Author):

Responses from the authors addressed most of the previous concerns and the quality of the revised manuscript is improved. Nevertheless, the authors still leave some major questions unresolved in the current version of manuscript.

Major points:

1. In the response letter, the authors agreed that "compared to our analysis of other stages, the screening experiments for stage-7 were not systematic but instead were done in multiple small experiments" and they also commented "LDN, T3, Repsox, GSIXX, RA, HGF, IGF1, and PD is all required for maintaining a high percentage of INS+ and NKX6.1+ cells at the end of stage 8". Given the focus of the paper, as indicated in the title, is "Novel chemical combinations potentiate human pluripotent stem cell-derived 3D pancreatic progenitor clusters toward functional β -cells", it is important to provide sufficient details and evidence to allow readers to see the data supporting the conclusions and be able to reproduce the differentiation method as reported by the authors. The revised manuscript is still lacking in data supporting stage 7 optimization and is only mentioned in one sentence (on Page 6 Line 163-165). Is it possible for the authors to provide data like Figure 2B or provide a table summarizing their optimizing trials to support their point that all the $i\beta$ -9C components are indeed necessary?
2. The authors wish to highlight that the novel combinations of several chemicals/factors are being reported for the first time in this manuscript. However, they didn't provide any mechanistic explanations and should delete the statement "discovered new mechanisms" in Line 50. The authors argued in the response letter that "they feel such a study is beyond the scope of the current work and better suited for a follow-up". However, at least in the Discussion section the authors should provide discussions on the potential mechanisms of the key chemicals they screened for the protocol and cite relevant previous studies to support their points. For example, the authors commented on FSK (a cAMP pathway activator) and CI-1033 (a pan-ErbB inhibitor) in promoting the stage 6 differentiation but they didn't give further explanations on if and how these pathways are related to beta cell function or differentiation. The same concerns are also applied to other "novel" compounds as highlighted in this manuscript, including GABA, HGF, IGF1, PD173074, G-1, Deza and ZM447439. If the space is limited in the manuscript, the authors can provide these information in supplementary materials.
3. Another remaining major concern is the lack of quantification and statistical analysis of data in main Figures 1-2 regarding protocol optimization and cell characterization. It is appreciated that the authors provided nice representative data (flow cytometry and immunostaining), however, they didn't show any statistics on these analyses. As a result, it is hard to appreciate the level of consistency or heterogeneity within the same differentiation and across different differentiations, especially for the Method-3 as developed by the authors in this report. Although the authors added new Supplementary Table 3 in this regard, why not just add quantification results and graphs in the main figures? For example, it is recommended to include quantification of results relevant to Figure 1G, 1I, 1M and 1O, and Figure 2B (at least the complete $\beta\beta$ -7C). In this way, readers can readily judge the advantages of the Method-3 protocol over previous Method-1/2 protocols.
4. Pertaining to the optimized differentiation protocol established in this study (as shown in Figure S2), can the authors comment on the rationale of the following compounds in terms of their temporal dosing manner?
 - λ RA: used on stage 3, stage 5-7 but not on stage 4 – Why do the authors think RA is not required at stage 4? Is there an explanation for dosing the RA with a gap at stage 4?
 - λ SANT1: used on stage 3, stage 5-6 – The same question described above is applied here.
 - λ ZnSO4: used on stage 5 and stage 7, but not on stage 6 or stage 8 – Is there an explanation for this intermittent dosing of zinc?

Minor points:

1. Page 9 Line 281 – Typos: H3149 10mg/ml should be 10ug/ml; ZnSO4 10mM should be 10uM.
2. Page 16 Line 529 – What is the abbreviation PP2 (5uM)? It wasn't mentioned elsewhere in this manuscript.
3. Figure S3A – The authors didn't mention the glucose concentration when applying exendin-4 in the static assay. Note, "exending-4" should be "exendin-4".
4. Line 374 the authors should correct the typo "merchodia"
5. Line 507 The authors indicate n=4 for all but one group but in Fig 3 the text shows n=3 for all groups – which is correct?
6. Fig 3 a& b, the authors should clarify what n's represent i.e. differentiation, technical reps, the authors should also provide the info about the islets use (ie ID, purity etc) in order for the reader to critically assess the quality of the islets being used to make comparisons.

Reviewer #2 (Remarks to the Author):

Revised study by Liu et al. answered many queries and provided a more comprehensive description on the Author's efforts to develop a robust protocol for human beta-like cell induction. Inclusion of human primary islets into GSIS assay as well as providing absolute values of Ins is of a big advantage of the revised manuscript. Similarly, provision of more detailed, precised, data on efficient subsequent step induction. It is a complex study and therefore there are several questions open. Though, this reviewer is aware it is not possible to address all of them in single publication. Few things still remain puzzling.

One of them is Activin A concentration at the first step of differentiation. How was the 115 ng/ml selected? Also, Authors claim that standard 100 ng/ml concentration might not sufficiently induce definitive endoderm? Can Authors elaborate on it? A concentration of 100 ng/ml in H1-hESC differentiation routinely yields more than 95% of cells expressing DE markers as was shown by numerous laboratories. Therefore, I am not sure what Authors mean "low-dose Activin-A did not efficiently induce hESCs into definitive endoderm at stage 1". Perhaps this reviewer missed it -is there any data included in the MS supporting this claim? One possible scenario is that cells at DE might robustly express DE markers, yet not represent fully specified endoderm. However, I think this should be clarified.

Reviewer #3 (Remarks to the Author):

The authors have response to most of the critiques.

However, in regard to point #9 - on transplantation of native islets, I feel this is a very important part of the paper since it would be a reference for future studies. Comparing the effects of the organoid-derived insulin-producing cells with native human islets would be highly relevant.

This Reviewer is still amazed that a blood glucose of 600 mg/dL drops down to less than 200 in a week!!

I disagree with the authors' view that because previous publications do not do show similar transplant studies this is going to be a new norm. If that is the case we will keep reducing experiments until we barely have any data !!

**Response to Reviewers' Comments:**

We thank the reviewers for recognizing our efforts and appreciating their additional suggestions. We
have addressed the comments from the reviewers in our revised manuscript and please see our point-
to-point responses (reviewers' comments are in *blue*).

**REVIEWER COMMENTS**

*Reviewer #1 (Remarks to the Author): Responses from the authors addressed most of the previous*
*concerns and the quality of the revised manuscript is improved. Nevertheless, the authors still leave*
*some major questions unresolved in the current version of manuscript.*

*Major points:*

*1. In the response letter, the authors agreed that “compared to our analysis of other stages, the*
*screening experiments for stage-7 were not systematic but instead were done in multiple small*
*experiments” and they also commented “LDN, T3, Repsox, GSIXX, RA, HGF, IGF1, and PD is all*
*required for maintaining a high percentage of INS+ and NKX6.1+ cells at the end of stage 8”. Given*
*the focus of the paper, as indicated in the title, is “Novel chemical combinations potentiate human*
*pluripotent stem cell-derived 3D pancreatic progenitor clusters toward functional β -cells “, it is*
*important to provide sufficient details and evidence to allow readers to see the data supporting the*
*conclusions and be able to reproduce the differentiation method as reported by the authors. The*
*revised manuscript is still lacking in data supporting stage 7 optimization and is only mentioned in*
*one sentence (on Page 6 Line 163-165). Is it possible for the authors to provide data like Figure 2B or*
*provide a table summarizing their optimizing trials to support their point that all the $i\beta$ -9C*
*components are indeed necessary?*

**Response:** We have added our optimizing strategy and flow cytometry data for stage-7 in
Supplementary Fig. 2E of the revised Figures.

*2. The authors wish to highlight that the novel combinations of several chemicals/factors are being*
*reported for the first time in this manuscript. However, they didn't provide any mechanistic*
*explanations and should delete the statement “discovered new mechanisms” in Line 50. The authors*
*argued in the response letter that “they feel such a study is beyond the scope of the current work and*
*better suited for a follow-up”. However, at least in the Discussion section the authors should provide*
*discussions on the potential mechanisms of the key chemicals they screened for the protocol and cite*
*relevant previous studies to support their points. For example, the authors commented on FSK (a*
*cAMP pathway activator) and CI-1033 (a pan-ErbB inhibitor) in promoting the stage 6 differentiation*
*but they didn't give further explanations on if and how these pathways are related to beta cell function*
*or differentiation. The same concerns are also applied to other “novel” compounds as highlighted in*
*this manuscript, including GABA, HGF, IGF1, PD173074, G-1, Deza and ZM447439. If the space is*
*limited in the manuscript, the authors can provide these information in supplementary materials.*

**Response:** We thank the reviewer for the suggestion and have added new content in the revised
manuscript to discuss the potential mechanism of four newly identified chemicals (FSK, ZM447439,
ISX-9 and G-1). These added sentences are: “Our study also reveals the importance of several
signaling pathway modulators in promoting the differentiation of hPSCs into β cells. Of these
modulators, (1) FSK has been reported to facilitate the process of epithelial-to-mesenchymal transition
(EMT) in several cell systems and thus might promote pancreatic progenitor into endocrine progenitor
at Stage-6 through inducing EMT (EMT is a critical step during endocrine progenitor specification)¹⁻⁴;
(2) ZM447439, an aurora kinase inhibitor, might promote terminal cell maturation status at Stage-8 by
facilitating cell cycle exit⁵⁻⁶; (3) ISX-9, might enhance hPSCs-derived β cells function and maintain
their identity at Stage-8 by inducing and maintaining NEUROD1 and INS expression in these cells⁷;

(4) G-1, a G protein-coupled estrogen receptor agonist, possibly promote β cells maturation at Stage-8
by modulating estrogen receptor related pathways as hPSCs-derived pancreatic progenitors mature
more quickly in female than male mice⁸. Further investigation is warranted to determine how exactly
these pathways exert their effects during the differentiation process.”

Reference:

- 1. Gouzi, M., Kim, Y.H., Katsumoto, K., Johansson, K. & Grapin-Botton, A. Neurogenin3 initiates
stepwise delamination of differentiating endocrine cells during pancreas development. *Dev Dyn* **240**, 589-
604 (2011).
- 2. Rukstalis, J.M. & Habener, J.F. Snail2, a mediator of epithelial-mesenchymal transitions,
expressed in progenitor cells of the developing endocrine pancreas. *GENE EXPRESSION PATTERNS* **7**, 471-479
(2007).
- 3. Cole, L., Anderson, M., Antin, P.B. & Limesand, S.W. One process for pancreatic beta-cell
coalescence into islets involves an epithelial-mesenchymal transition. *J ENDOCRINOL* **203**, 19-31 (2009).
- 4. Xue, Y., Sun, R., Zheng, W., Yang, L. & An, R. Forskolin promotes vasculogenic mimicry and
invasion via Notch1-activated epithelial-mesenchymal transition in syncytiolization of trophoblast cells in
choriocarcinoma. *INT J ONCOL* **56**, 1129-1139 (2020).
- 5. Willems, E. *et al.* The functional diversity of Aurora kinases: a comprehensive review. *CELL DIV*
**13**, 7 (2018).
- 6. Myster, D.L. and R.J. Duronio, To differentiate or not to differentiate? *Curr Biol*, 2000. 10(8): p.
R302-4.
- 7. Dioum, E.M. *et al.* A small molecule differentiation inducer increases insulin production by
pancreatic cells. *Proceedings of the National Academy of Sciences* **108**, 20713-20718 (2011).
- 8. Saber, N. *et al.* Sex Differences in Maturation of Human Embryonic Stem Cell-Derived beta Cells
in Mice. *ENDOCRINOLOGY* **159**, 1827-1841 (2018).

*3. Another remaining major concern is the lack of quantification and statistical analysis of data in*
*main Figures 1-2 regarding protocol optimization and cell characterization. It is appreciated that the*
*authors provided nice representative data (flow cytometry and immunostaining), however, they didn't*
*show any statistics on these analyses. As a result, it is hard to appreciate the level of consistency or*
*heterogeneity within the same differentiation and across different differentiations, especially for the*
*Method-3 as developed by the authors in this report. Although the authors added new Supplementary*
*Table 3 in this regard, why not just add quantification results and graphs in the main figures? For*
*example, it is recommended to include quantification of results relevant to Figure 1G, 1I, 1M and 1O,*
*and Figure 2B (at least the complete F β -7C). In this way, readers can readily judge the advantages of*
*the Method-3 protocol over previous Method-1/2 protocols.*

**Response:** We have added statistical analysis data in the Main Text of the manuscript for information
related to figure 1G, 1I, 1M and 1O. For F β -7C of Figure 2B, as we previously showed the statistical
analysis data in the manuscript, we highlighted it in yellow in the revised manuscript. For all relevant
raw data, we have put them into the new Source Data file.

*4. Pertaining to the optimized differentiation protocol established in this study (as shown in Figure*
*S2), can the authors comment on the rationale of the following compounds in terms of their temporal*
*dosing manner?*

*RA: used on stage 3, stage 5-7 but not on stage 4 – Why do the authors think RA is not required at*
*stage 4? Is there an explanation for dosing the RA with a gap at stage 4?*

**Response:** We determined the temporal dosing manner of each compound based on the screening
results. We found that adding high concentration of RA (2 μ M) at stage-3 is critical for inducing

PDX1 expression. However, for stage-4, we discovered that adding RA blocks the expression of
NKX6.1, so we removed RA from stage-4. In fact, using RA on Stage-3 and omitting it on Stage-4 is
also implemented in most of published protocols for generation of beta cells from hPSCs. Another
group examined the effect of RA signaling in the pancreatic development in the context of zebrafish
embryogenesis (Huang, W., et al.). This study concluded that RA plays a biphasic role in pancreas
development: RA initially patterns the endoderm and specifies the pancreatic field, but later
negatively regulates the further differentiation of pancreatic progenitors. At Stage 5-7, our screening
results showed that adding low concentration of RA (0.05 μ M) increases the production of beta cells
in the end.

Huang, W., et al., Retinoic acid plays an evolutionarily conserved and biphasic role in pancreas
development. Dev Biol, 2014. 394(1): p. 83-93.

*SANT1: used on stage 3, stage 5-6 – The same question described above is applied here.*

**Response:** Similar to the case of RA, our screening results showed that adding SANT1 (a hedgehog
pathway inhibitor) at this stage 5-6 (but not stage-4) will improve the final beta cell production. We do
not know the exact mechanism for this temporal dosing of SANT1. A possible explanation is that the
hedgehog pathway activity needs to be carefully modulated temporally during the process of beta cell
generation.

*ZnSO4: used on stage 5 and stage 7, but not on stage 6 or stage 8 – Is there an explanation for this*
*intermittent dosing of zinc?*

**Response:** Similar to the temporal dosing manner of RA and SANT1, we found that ZnSO4 benefits
stages 5 and 7 but not stages 6 or 8.

*Minor points:*

*1. Page 9 Line 281 – Typos: H3149 10mg/ml should be 10ug/ml; ZnSO4 10mM should be 10uM*

**Response:** We thank the reviewer for pointing these out and have corrected the typos in the revised
manuscript.

*2. Page 16 Line 529 – What is the abbreviation PP2 (5uM)? It wasn't mentioned elsewhere in this*
*manuscript.*

**Response:** PP2 is pyrrolo-pyrimidine Src family kinase inhibitor, which has been used to induce
NGN3 expression from pancreatic progenitor (Ivka Afrikanova et al., Inhibitors of Src and Focal
Adhesion Kinase Promote Endocrine Specification Impact on the Derivation of beta-cells from
Human Pluripotent Stem Cells. 2011, The Journal of Biological Chemistry 286, 36042-36052). We
have added the information for PP2 in the revised manuscript.

*3. Figure S3A – The authors didn't mention the glucose concentration when applying exendin-4 in the*
*static assay. Note, "exending-4" should be "exendin-4".*

**Response:** Exendin-4 was administrated with basal level of glucose, which is 3.3mM. We had added
this information in the figure legend of the revised manuscript. We have corrected this typo in the
revised manuscript.

*4. Line 374 the authors should correct the typo “merchodia”*

**Response:** We have corrected this typo in the revised manuscript.

*5. Line 507 The authors indicate n=4 for all but one group but in Fig 3 the text shows n=3 for all*
*groups – which is correct?*

**Response:** There seems to be some misunderstandings here. In the figure legend, “n=4 for the
experimental group, and n = 3 for the control group” refers to Figure 3C. While in the text of Figure 3,
“n=3 for all groups” only refers to Figure 3D. These are two different figure legends.

*6. Fig 3 a& b, the authors should clarify what n’s represent i.e. differentiation, technical reps, the*
*authors should also provide the info about the islets use (ie ID, purity etc) in order for the reader to*
*critically assess the quality of the islets being used to make comparisons.*

**Response:** We have added this information in the figure legend of our revised manuscript.

*Reviewer #2 (Remarks to the Author):*

*Revised study by Liu et al. answered many queries and provided a more comprehensive description on*
*the Author's efforts to develop a robust protocol for human beta-like cell induction. Inclusion of*
*human primary islets into GSIS assay as well as providing absolute values of Ins is of a big advantage*
*of the revised manuscript. Similarly, provision of more detailed, precised, data on efficient subsequent*
*step induction. It is a complex study and therefore there are several questions open. Though, this*
*reviewer is aware it is not possible to address all of them in single publication. Few things still remain*
*puzzling.*

*One of them is Activin A concentration at the first step of differentiation. How was the 115 ng/ml*
*selected? Also, Authors claim that standard 100 ng/ml concentration might not sufficiently induce*
*definitive endoderm? Can Authors elaborate on it? A concentration of 100 ng/ml in H1-hESC*
*differentiation routinely yields more then 95% of cells expressing DE markers as was shown by*
*numerous laboratories. Therefore, I am not sure what Authors mean “low-dose Activin-A did not*
*efficiently induce hESCs into definitive endoderm at stage 1”. Perhaps this reviewer missed it -is there*
*any data included in the MS supporting this claim? One possible scenario is that cells at DE might*
*robustly express DE markers, yet not represent fully specified endoderm. However, I think this should*
*be clarified.*

**Response:** At the beginning, we tested Activin-A at 100 ng/ml for 3 days on hPSCs with cell density
around 70-80% confluence (a confluence used in several published studies) and found that it could
efficiently induce hPSCs into definitive endoderm (>90%, based on the expression of markers FOXA2
and SOX17). However, with such a cell density, we failed to generate high percentage of pancreatic
progenitors at Stage-4. By further analysis, we discovered that NKX6.1 expression at Stage-4 mainly

showed up in areas with high cell density. In addition, when the overall density of the cell culture was
very low at stage-1, very low NKX6.1 expression will be induced at stage-4 (despite the PDX1
expression was not greatly affected). Therefore, we increased the starting hES cell density to about 90%
confluence. In such a higher cell density, when 100 ng/ml Activin A was applied for three days at
stage-1, we frequently observed a certain amount of undifferentiated hPSCs (~10%-25%) at the end of
stage-1. To identify the optimal Activin A concentration, we then tested different concentrations of
Activin A for stage-1, including 80, 90, 95, 100, 105, 110, 115, 120, 125, 130 and 140 ng/ml, and
finally found 115 ng/ml worked as the best for day-1 in our hand. Therefore, we used gradient Activin
A in our final protocol, which were 115ng/ml (day1), 110ng/ml (day2) and 100ng/ml (day3) as
described in Supplementary Fig.1E. In sum, one possible reason for the use of slightly higher Activin
A than most published protocols is that higher starting hPSCs cell density can generate higher
NKX6.1 expression at Stage-4, but also requires a higher dose of Activin A to initiate the
differentiation.

As you can see in Supplementary Fig. 1E, constant low-dose Activin-A (100ng/ml for three days)
used at stage 1 resulted in a significant portion of NKX6.1-negative cells at stage 4. To avoid the
confusion, we have rewritten this sentence to “constant low-dose Activin-A generated cell culture
with appropriate cell density but resulting in high cell heterogeneity at stage 4 (Supplementary
Fig.1E).”

*Reviewer #3 (Remarks to the Author):*

*The authors have response to most of the critiques. However, in regard to point #9 - on transplantation*
*of native islets, I feel this is a very important part of the paper since it would be a reference for future*
*studies. Comparing the effects of the organoid-derived insulin-producing cells with native human*
*islets would be highly relevant. This Reviewer is still amazed that a blood glucose of 600 mg/dL drops*
*down to less than 200 in a week!!*

**Response:** In our experiments, diabetic mice that were transplanted with stage-8 cells reverted to
normoglycemia (<250 mg/dL) within 2 weeks, not as quick as “less than 200 in a week”. Even after 3
200 weeks, some mice still remained glucose levels a litter higher than 200 mg/dL (please see our
supplementary Source Data file and Figure 3C). Such “fast reversal” is reasonable as beta-like cells
partially reprogrammed from human alpha cells could reverse diabetes in mice model less than 7 days
(see Figure 2C in “Furuyama, K., et al., Nature, 2019. 567(7746): p. 43-48.”). Such similar “fast
reversal” was also reported in another literature using hPSCs-derived beta cells (Vegas, A.J., et al.,
Nature Medicine, 2016. 22(3): p. 306-311).

*I disagree with the authors' view that because previous publications do not do show similar transplant*
*studies this is going to be a new norm. If that is the case we will keep reducing experiments until we*
*barely have any data !!*

**Response:** We agree with the reviewer that more control data will lend more support to our findings.
Therefore, we did do our best to include adult human islets in our *in vitro* studies of both the glucose
stimulated C-peptide secretion (Figure 3A) and the measurements of total insulin content (Figure 3B)
assays. However, due to the current COVID19 impact, many institutes including ours have strict
restrictions in multiple on-campus activities, especially those concerning animal experiments. In
parallel, based on our review of the recent literatures, we do notice that several important publications
on hESC-differentiated beta cells published in high profile journals, including Vegas, A. et al., Nature
Medicine, 2016; Ghazizadeh et al., Nature Communication, 2017; Rosado-Olivieri et al., Nature
Communication, 2019, did not provide primary human islets as such in vivo control, and just

presented the critical comparison between transplanted hPSC-derived beta cells and the no treatment
control. In these cases, only one control was sufficient to support their conclusions. In addition to the
no treatment control, we also provided the transplantation of fibroblasts as another control group.
Therefore, from the point of view to provide sufficient information, we did not reduce the number of
experiments or lower the bar. To better present our efforts in these experiments, we summarized our
works and raw data into the new Source Data file.

REVIEWERS' COMMENTS

Reviewer #1 (Remarks to the Author):

The authors have addressed all previous concerns and the revised manuscript provides a more detailed description. Only few minor points are suggested to address:

1. Supplementary Figure 1H – Ideally the DAPI channel contrast should be consistent between images.
2. Supplementary Figure 2E – The authors have added more information about the optimization trials for stage 7. From the experimental design illustration shown in Figure S2E, it seems the authors optimized stage 7 conditions separately rather than based on or followed by their identified optimal stage 6 conditions (“EP-8C”, Figure S2C). So this reviewer assumes that the authors optimized stage 8 conditions separately as well without being based on their identified optimal stage 6/7 conditions. Can the authors clarify this confusion or comment on the rationale of optimizing each stage separately in this study?
3. The authors should include the Actvin A data that are in the rebuttal, in the supplement as it would strengthen the manuscript, especially since the AA conc is important to DE formation.
4. The bi-phasic addition of certain small molecules is interesting (RA, SANT1, ZnSO₄). I think the authors should draw attention to this in their manuscript as they have in their rebuttal. The benefits of ZnSO₄ at only S5 and S7 is intriguing - does ZnSO₄ negatively impact the expression of these genes at S6 and S8? Or did the authors find no difference in insulin and NKX6.1 expression with or without ZnSO₄ at S6 and S8, so decided to remove it. “We found that ZnSO₄ benefits S5 and 7...”, how does it benefit S5 exactly? Increased NGN3 and NEUROD1? Increased INS? Enhanced cell survival?
5. the authors should clearly state what beta cell efficiency means. Is it INS+ or INS+/NKX6.1+?
6. Figure 3B – It is recommended that the color selection for stage-8 cells and primary human islets Figure 3B be kept consistent with that used in Figure 3A (green box for stage-8 cells and black triangle for primary human islets). In addition, the total insulin content measured for primary human islets seems significantly lower than most of previous reports (as far as this reviewer knows, primary human islet contain 15-30ng per IEQ or per 1000 cells). Nevertheless, the total insulin content measured for stage-8 cells is still an impressive value compared with many previous reports. The authors should also include descriptions about how they extract total insulin content in the related Methods (Page 11 Line 385-393).
7. Page 10 Line 313 and Line326 – Typos: Heparin 10mg/ml should be 10ug/ml.
8. The authors should reorder or rename the images in Fig S2. Currently, it is arranged as a, b, d, f, c, e.
9. Line 413, the authors should modify the citation to match the appropriate format.

REVIEWERS' COMMENTS

Reviewer #1 (Remarks to the Author):

The authors have addressed all previous concerns and the revised manuscript provides a more detailed description. Only few minor points are suggested to address:

Response: We thank the review for finding our revisions satisfactory and the further minor comments.

1. Supplementary Figure 1H - Ideally the DAPI channel contrast should be consistent between images.

Response: According to reviewer's suggestion, we have modified the contrast as well as brightness to make the DAPI channel consistent between images.

2. Supplementary Figure 2E - The authors have added more information about the optimization trials for stage 7. From the experimental design illustration shown in Figure S2E, it seems the authors optimized stage 7 conditions separately rather than based on or followed by their identified optimal stage 6 conditions ("EP-8C", Figure S2C). So this reviewer assumes that the authors optimized stage 8 conditions separately as well without being based on their identified optimal stage 6/7 conditions. Can the authors clarify this confusion or comment on the rationale of optimizing each stage separately in this study?

Response: Yes, at the beginning we optimized the stage 6, 7, and 8 separately. The reason for this strategy is that the β cell differentiation from hPSCs is a multiple-step time-consuming protocol; therefore it would be more efficient to optimize each stage separately (rather than one after another). In this way, we can optimize all stages simultaneously. For instance, we did not wait until we find the optimal condition for stage 6 before we optimize stage 7 and stage 8. In this way, we can speed up the whole optimization process. After we are done with the optimization at each stage, we picked the top three conditions at each stage and further test the combinations of them. For instance, when we started to optimize the conditions for stage 7, we had not got the optimal condition for stage 6, so we used a sub-optimal condition of stage 6 to optimize stage 7. We then further test all combinations of the top three conditions of stage 6, 7, and 8, and select the best combination. In this manner, we achieve our final differentiation recipes for stage 6, 7, and 8.

3. The authors should include the Actvin A data that are in the rebuttal, in the supplement as it would strengthen the manuscript, especially since the AA conc is important to DE formation.

Response: According to reviewer's suggestion, we have added those descriptions from the previous rebuttal to the supplementary method to strengthen the manuscript.

4. The bi-phasic addition of certain small molecules is interesting (RA, SANT1, ZnSO₄). I think the authors should draw attention to this in their manuscript as they have in their rebuttal. The benefits of ZnSO₄ at only S5 and S7 is intriguing - does ZnSO₄ negatively impact the expression of these genes at S6 and S8? Or did the authors find no difference in insulin and NKX6.1 expression with or without ZnSO₄ at S6 and S8, so decided to remove it.

“We found that ZnSO₄ benefits S5 and 7...”, how does it benefit S5 exactly? Increased NGN3 and NEUROD1? Increased INS? Enhanced cell survival?

Response: We found adding ZnSO₄ at stage 5 (as well as stage 7) will slightly improve the NKX6.1+/INS+ cell percentage and the glucose-stimulated c-peptide secretion function of the beta cells when we checked the cells at stage 8. We did not observe a significant change after adding ZnSO₄ at stage-6 or stage-8, so we did not include it into our optimized protocol. For stage 8, the supplementary element mix (Trace Elements A (2000x, Corning, MT99182CI)) contains ZnSO₄. Although the concentration is undisclosed, it might be possible that Zn ion from this source is already sufficient.

5. The authors should clearly state what beta cell efficiency means. Is it INS+ or INS+/NKX6.1+?

Response: We used percentage of INS+/NKX6.1+ cells as the readout of the beta cell efficiency throughout the manuscript to exclude the contribution of some bi-hormonal INS+/GCG+ cells. This is also a common practice in the field as it is clear that INSULIN cannot stand alone as the marker of beta cells. We have clarified this in the updated abstract.

6. Figure 3B – It is recommended that the color selection for stage-8 cells and primary human islets Figure 3B be kept consistent with that used in Figure 3A (green box for stage-8 cells and black triangle for primary human islets).

Response: According to the reviewer's suggestion, we have changed the colors in Figure 3B to make it consistent with Figure 3A. .

In addition, the total insulin content measured for primary human islets seems significantly lower than most of previous reports (as far as this reviewer knows, primary human islet contain 15-30ng per IEQ or per 1000 cells). Nevertheless, the total insulin content measured for stage-8 cells is still an impressive value compared with many previous reports. The authors should also include descriptions about how they extract total insulin content in the related Methods (Page 11 Line 385-393).

*Response: Each primary human islet has been reported to contain as low as 4ng insulin to as high as ~60ng (Henquin, J.C., Mol Metab, 2019. 30: p. 230-239.). Based on this most recent and thorough review article, generally speaking, primary human islet contains 13.3 ng/IEQ on average. While based on another thorough study (Pisania, A., et al., Lab Invest, 2010. 90(11): p. 1661-75.), each IEQ contains about 1560 cells. Thus 1000 cell has 13.3ng/1.560 equals to 8.52ng insulin content. In addition, we noticed that a study by Dr. Melton's group published in Cell stated that insulin content is “200 ±40μIU/1000 cells” for human primary islets (Pagliuca, F.W., et al. Cell, 2014. 159(2): p. 428-439.). The conversion factor for human insulin is: One International Unit (IU) equals 0.0347 mg of insulin. So islets has $(200 \pm 40) * 0.0347 / 1000,000 * 1000,000 = 6.94 \pm 1.39 \text{ng insulin /1000 cells}$. Therefore, our data, which is about 7ng/1000 cells, is comparable to the values reported by others.*

We have included descriptions about how we extracted total insulin content in the related Methods.

7. Page 10 Line 313 and Line326 – Typos: Heparin 10mg/ml should be 10ug/ml.

Response: We have corrected this typo.

8. The authors should reorder or rename the images in Fig S2. Currently, it is arranged as a, b, d, f, c, e.

Response: we have noticed this from the very beginning. However, Fig.S2 (now Fig. S3) is almost 100% full, and we do not have any extra space as panel c and e are a bit big. The current order looks a little awkward. But if we name the images in Fig S2 orderly as a-b-c-d-e-f, we will cite these images in the manuscript disorderly like a-b-d-f-c-e. We think the current arrangement is the better one of the two “bad” solutions.

9. Line 413, the authors should modify the citation to match the appropriate format.

Response: We have corrected this formatting issue.